## BRIEF COMMUNICATION
# Searching thousands of genomes to classify somatic and novel structural variants using STIX

Murad Chowdhury[1], Brent S. Pedersen[2], Fritz J. Sedlazeck [3], Aaron R. Quinlan[4,5,6] and Ryan M. Layer [1,7 ✉]

**Structural variants are associated with cancers and developmental disorders, but challenges with estimating population frequency remain a barrier to prioritizing mutations over inherited variants. In particular, variability in variant calling heuristics and filtering limits the use of current structural variant catalogs. We present STIX, a method that, instead of relying on variant calls, indexes and searches the raw alignments from thousands of samples to enable more comprehensive allele frequency estimation.**

Structural variants (SVs), including large deletions, duplications, insertions, inversions and translocations[1], are associated with cancer progression and Mendelian disorders[2–5]. Copy number variants and gene fusions have received the most attention, but recent large-scale SV studies such as the Pan-Cancer Analysis of Whole Genomes[6] (PCAWG), the 1,000 Genome Project[7] (1KG), gnomAD SV[8] and the Centers for Common Disease Genomics[9] (CCDG) have expanded our understanding of the depth and diversity of somatic SVs in cancer and polymorphic SVs in humans. Despite the importance of SVs, barriers remain to their adoption in disease analysis[1]. In particular, limitations to short-read SV calling, reference biases and variability in the heuristics and filtering strategies between cohorts lead to an incomplete understanding of SV population frequency that limits our ability to assess a variant's severity and impact[10].

In cancer studies, SV interpretation requires classifying variants as germline or somatic. The standard is to call variants in the tumor and control tissue from the same individual. SVs found only in the tumor are deemed somatic. This method is susceptible to the sensitivity of the normal sample calls, which are often sequenced at lower coverage. When an inherited SV is missed in the normal tissue, it can be incorrectly classified as somatic. An alternative strategy is to substitute matched-normal tissue with a panel of unrelated normal samples (for example, 1KG, Simons Genome Diversity Panel[11] (SGDP)), but the time and computational costs associated with joint calling large numbers of samples can be prohibitively high.

SV catalogs from large DNA sequencing projects can filter tumor-only calls as a shortcut to joint calling. Variants found in both the tumor and reference catalog can be classified as inherited since we can reasonably assume that somatic variants, and driver mutations in particular, are likely to be rare and unlikely to share SV breakpoints with polymorphic SVs. While this assumption does not hold in all cases, it is the standard for many diseases studies. The analysis is more complicated for variants found only in the tumor calls. In principle, SVs that are not in the cohort are rare and could be somatic. In practice, several SV-specific factors,

including short-read calling limitations[12], genotyping complexities (Supplementary Note 1 and Supplementary Fig. 1) and aggressive filtering for false-positive calls, exclude many real SVs from appearing in these catalogs. For example, among the thousands of cancer-related SVs that are recoverable in 1KG, fewer than 500 are present in the 1KG SV call set[7]. Given these issues, it is impossible to determine whether an SV observed in a patient and not in a reference cohort is absent from the population (true negative) or removed in the filtering step (false negative).

Similarly, in Mendelian disease analyses, causal variants should be either absent or are at very low-frequency in the reference population[13]. Using allele frequencies from gnomAD[14], a catalog of single nucleotide variants from 141,546 human genomes, can reduce the number of variants under diagnostic consideration by two orders of magnitude[13]. Unfortunately, no equivalent resource exists for SVs since, as with the cancer analysis, static call sets from large populations are inadequate. Pangenomes can help by identifying and genotype SVs[15], but given the limited number of samples and SVs they can currently represent, they are better suited to common variants and are less useful for somatic and pathogenic variant classification.

To ensure comprehensive and accurate SV detection and allele frequency assignment, we propose searching the raw alignments across thousands of samples using our SV index (STIX). For a given deletion, duplication, inversion or translocation, STIX reports a per-sample count of every alignment that supports the variant (Fig. 1). Assuming deleterious variants are rare, from these counts, we can conclude that an SV with evidence in many healthy samples is either a common germline variant or the product of systematic noise (for example, an alignment artifact) and is unlikely to be pathogenic. By relying on the raw alignments, STIX avoids the previously described false negative issues and removes thousands of variants that could have otherwise been associated with disease.

STIX is built on top of the GIGGLE genome search engine[16]. Sequence alignment files contain mostly normal alignments and typically less than 5% 'discordant' alignments (split reads and paired-end reads with unexpected aligned distance between pairs or strand configuration) due to either the presence of a SV or some noise in the sequencing or alignment process (Fig. 1a). These alignment signals are used for detection by all current methods. STIX extracts and tracks all discordant alignments from each sample's genome (Fig. 1a), then creates a unified GIGGLE index for all samples. When a user provides the SV type, breakpoint coordinates and its confidence intervals, STIX returns the count of all alignments that support the variant (Fig. 1b). We have deployed web interfaces

[1]BioFrontiers Institute, University of Colorado, Boulder, CO, USA. [2]University Medical Center, Utrecht University, Utrecht, the Netherlands. [3]Human Genome Sequencing Center, Baylor College of Medicine, Houston, TX, USA. [4]Department of Human Genetics, University of Utah, Salt Lake City, UT, USA. [5]Department of Biomedical Informatics, University of Utah, Salt Lake City, UT, USA. [6]Utah Center for Genetic Discovery, University of Utah, Salt Lake City, UT, USA. [7]Department of Computer Science, University of Colorado, Boulder, CO, USA. ✉e-mail: ryan.layer@colorado.edu

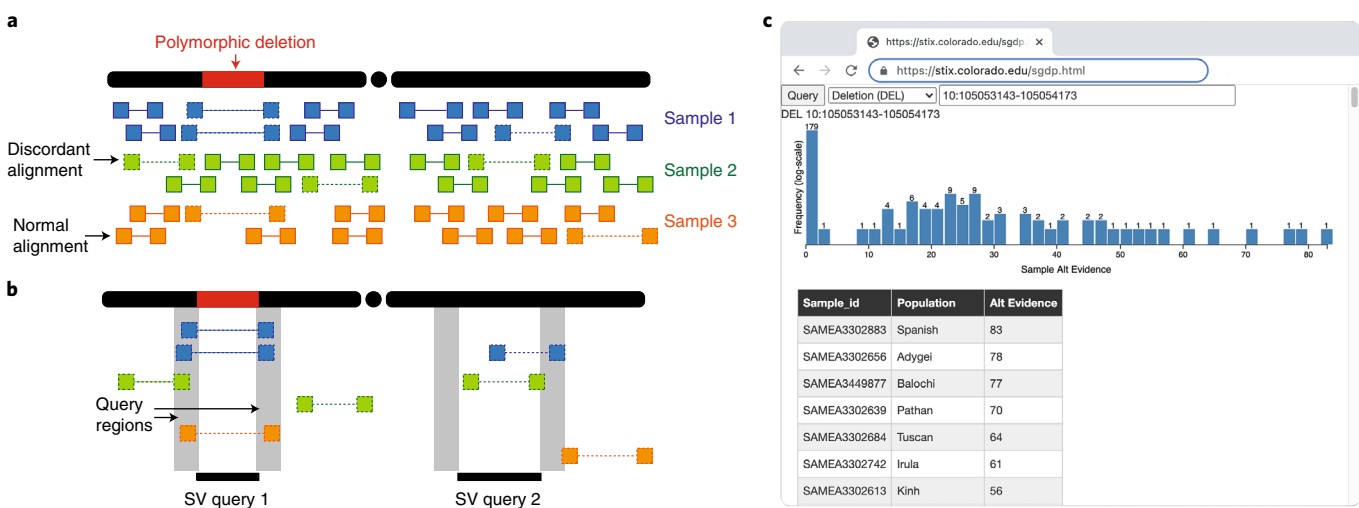

**Fig. 1 | The STIX SV index. a,b,** The STIX indexing and query process for three samples and a polymorphic deletion. **a,** A small number of the alignments that tile the genomes are discordant (designated by a dotted line connected read pairs) because of either an SV or other nonspecific causes (for example, mapping artifacts). **b,** Discordant alignments are extracted from all samples and indexed using GIGGLE. Query SVs are mapped to alignments that reside in both regions and are aggregated and returned. The first query returns three alignments in two samples and the second returns zero alignments. **c,** The distribution of evidence depths for a deletion searched in the SGDP cohort through the http://stix.colorado.edu interface.

for STIX queries of 1KG and SGDP aligned to GRCh37 at http://stix.colorado.edu (Fig. 1c). The server also supports direct access for integrating STIX into other programs.

Considering the 1KG SV catalog, STIX shows high accuracy in identifying the samples with deletions (0.998), duplications (0.995) and inversions (0.988) (Methods and Supplementary Table 1). This result is consistent with a previous report showing STIX outperformed DELLY, SVTyper and SV2 on simulated and real deletions, and demonstrated the best balance between sensitivity and specificity[17]. The STIX index was also 500× smaller than the original alignments and queries ran 620× faster (Methods).

Using STIX indexes of 1KG and SGDP, we recovered thousands of somatic SVs published in the Catalogue of Somatic Mutations in Cancer[18] (COSMIC) and PCAWG (Fig. 2a,b,d,e). These variants were likely either germline or recurrent mutations and unlikely to be driving tumor evolution. Only a fraction of the SVs found by STIX were in either the 1KG or gnomAD SV lists (Fig. 2c,f) (Supplementary Note 2).

STIX's primary use is to refine SV calls down to a set that can be assessed manually, especially in the absence of DNA sequences from matched-normal tissue. For example, when applied to 183 prostate cancer samples from PCAWG, the MANTA[19] caller recovered, on average, 3,892.8 deletions per sample (Fig. 2g). Using the PCAWG calls as the truth set, removing SVs using the matched-normal tissue resulted in 51.4 false positives, 29.9 true positives and 3.7 false negatives. Using the STIX 1KG database had 30% fewer false positives (35.8), roughly the same number of true positives (23.1) and some additional false negatives (10.5). The results were similar for inversions (Supplementary Fig. 2a) and duplications (Supplementary Fig. 2b). In addition to being over 50× smaller than the tumor-only call set, the STIX-filtered calls were also enriched for putatively consequential variants (Supplementary Fig. 3). Using the 1KG and gnomAD SV calls as germline filters was less effective because the average number of false positives was 88× and 55× higher, respectively. These population filters' true-positive and false-positive results were similar to the STIX results, indicating that the PCAWG call set likely retained some common SVs.

STIX enables fast and accurate SV frequency estimates directly from population-scale sequencing data, which was not possible in previous SV studies due to inconsistent filtering and calling strategies. It does this by indexing all SV evidence directly from the raw alignments, avoiding detection bias, and compressing large consortia data sets. With STIX, we indexed 2,504 samples from the 1,000 Genomes Project and 279 samples from the Simons Genome Diversity Project. These indexes helped improve somatic SV calls and highlighted the potential for recurrent de novo SVs (Supplementary Note 3). The code is freely available at https://github.com/ryanlayer/stix.

A limitation of this approach is that, while population frequency is a powerful metric for isolating rare, potentially functional variants, not all rare variants are pathogenic, and making this classification requires further analysis. Additionally, with STIX, and all alignment-based short-read SV methods, it is difficult to determine whether two discordant alignments support the same SV or similar SVs. Discordant paired-end reads provide indirect evidence of an SV, which leads to breakpoint location ambiguity that can affect STIX's resolution (Supplementary Note 4). STIX also does not track per-sample normal coverage levels (due to high storage cost) and cannot distinguish between no support for an SV and insufficient coverage at a particular locus. When considering a large reference cohort, coverage fluctuations in individual samples minimally affect the results. Quantifying read depth or applying other quality control metrics is advisable for smaller cohorts or particularly sensitive experiments involving rare variants.

In the future, we plan to explore how STIX may enable data access with lower consent and privacy issues. Reporting summary statistics reduces the likelihood of re-identifying samples, which would help reconcile different consent rights across patient cohorts. With these improvements, STIX could bring the power of thousands of genomes to the diagnosis and treatment decisions process.

## Online content

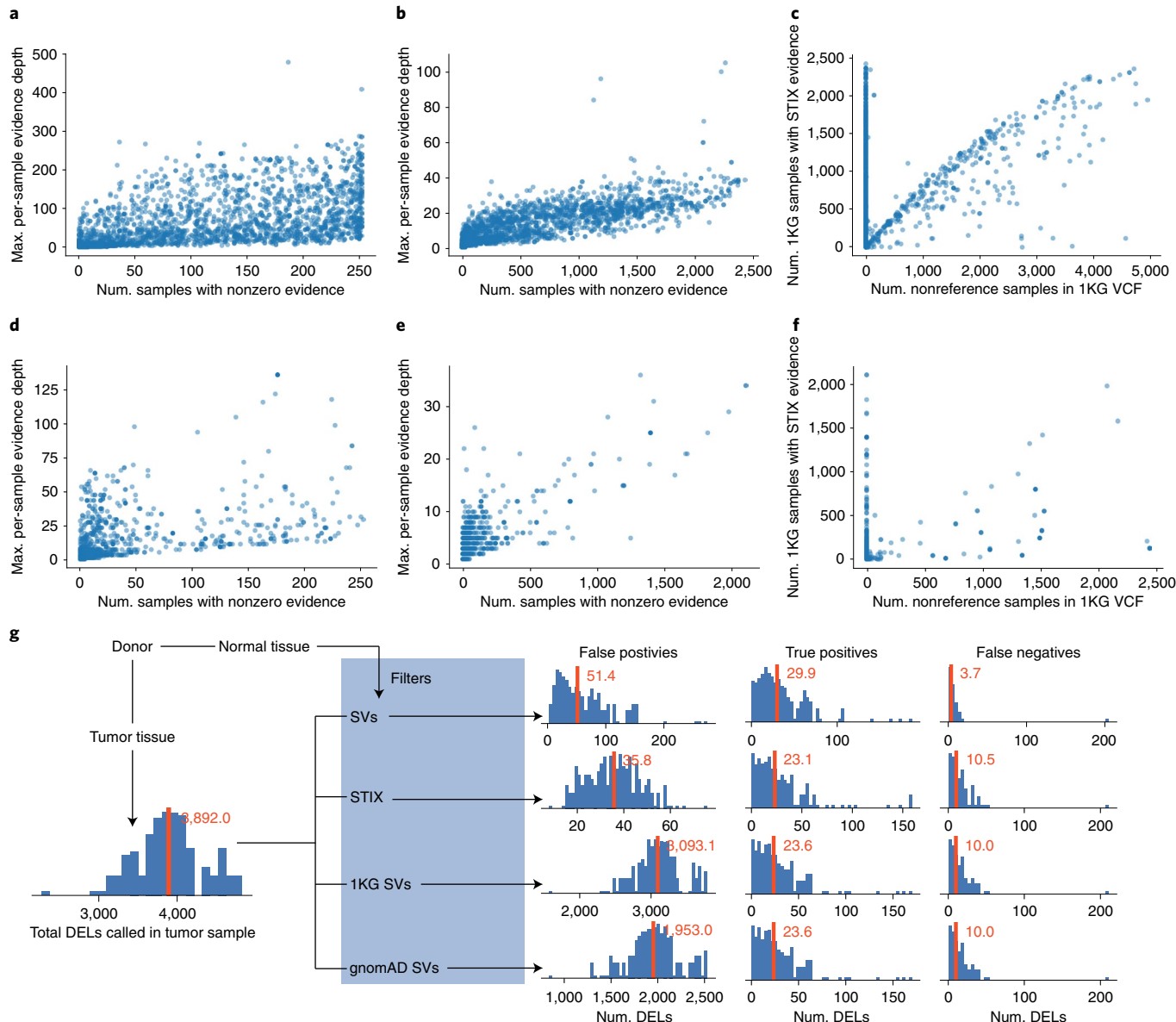

**Fig. 2 | SVs reported in cancer databases also occur in healthy populations. a–c**, COSMIC contains 46,185 somatic deletions. STIX found evidence for 27.9% of these SVs in SGDP (**a**) and 27.5% in 1KG (**b**). In these two plots (and **d** and **e**), we summarize the population-level evidence for each recurring SV (blue dot) by the number of (Num.) samples with any concordant evidence (x axis) and the maximum (Max.) amount of per-sample evidence (y axis). **c**, Only 1% of COSMC SVs appeared in the 1KG SV call set. The agreement between the STIX and the 1KG call sets is plotted using the population frequency estimates from each method for each SV. **d–f**, PCAWG found 84,083 deletions, 3.4% of which were in SGDP (**d**) and 2% were in 1KG (**e**). **f**, The 1KG call set contained only 0.2% PCAWG SVs. **g**, A comparison of germline filtering strategies for 183 prostate tumor samples that remove tumor deletions found in matched-normal tissue (SV), the STIX index of 1KG, the 1KG SV calls and the gnomAD SV calls. Histograms show the frequency of sample-level SV counts. Red bars and text give the sample mean. For example, the raw tumor calls had, on average, 3,892.0 SVs and STIX filtering yielded, on average, 35.8 false positives, 23.1 true positives and 10.5 false negatives.

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

## Methods

**STIX SV evidence collection and classification.** When a user submits a query, they specify the SV type (deletion, duplication, inversion or break end) and breakpoint. Breakpoints are encoded as a pair of left and right coordinates, where each coordinate has a chromosome and start and end positions. The left coordinate is strictly downstream of the right and has a lexicographically equivalent or smaller chromosome. The left and right coordinates are extended to account for the indexed samples' insert size distribution and the SV type. Deletions extend the left coordinate downstream and the right coordinate upstream. Duplications extend the left coordinate upstream and the right coordinate downstream. Inversions extend both coordinates downstream for + strand alignments and upstream for − strand alignments. For break ends, the left and right coordinates are not modified.

STIX searches the index using the left coordinate and only retains alignments that also overlap the right coordinate and have a strand configuration that matches the given SV type (listed below). STIX counts the number matches per sample and reports that total to the user.

SV strand configurations:

**Deletions:**

- Paired-end alignments must have a +/− orientation.
- Split-read alignments must have a +/+ or −/− orientation.

**Duplications:**

- Paired-end alignments must have a −/+ orientation.
- Split-read alignments must have a +/+ or −/− orientation.

**Inversions:**

- Paired-end alignments must have a +/+ or −/− orientation.
- Split-read alignments must have a +/− or −/+ orientation.

**Break ends:**

- The only requirement is that the alignments overlap the left and right query coordinates.

**SV evidence extraction and STIX index creation.** SV alignment evidence (discordant reads and split reads) are extracted from BAM and CRAM files using excord (https://github.com/brentp/excord). Excord scans each alignment to determine whether it contains a split read, has a strand configuration that is not +/−, the two aligned ends are not on the same chromosome and the distance between the two aligned ends is further away than expected (set by the --discordantdistance command line parameter). The expected distance between two reads depends on the size and variance of fragments and can be measured by finding the mean and standard deviation of normal alignments in the BAM file. We recommend using the mean plus two times the standard deviation for the discordant distance. If any of these conditions is true, then the alignment is recorded as a possible piece of SV evidence. For each piece of evidence, excord stores the position and orientation of the two ends into a sample-specific BED file. For example,

| 1 | 10022 | 10122 | 1 | 1 | 249240455 | 249240538 | 1 | 0 |
|---|-------|-------|---|---|-----------|-----------|---|---|
| 1 | 10031 | 10131 | 1 | 4 | 191044177 | 191044238 | 1 | 0 |
| 1 | 10036 | 10136 | 1 | 2 | 243153001 | 243153102 | −1 | 0 |
| 1 | 10054 | 10154 | −1 | 19 | 59097998 | 59098033 | −1 | 0 |
| 1 | 10066 | 10166 | −1 | 1 | 249239980 | 249240049 | −1 | 0 |

Excord was written in the Go programming language. Precompiled binaries are available under releases in its GitHub repository.

Each sample BED file is sorted and bgziped. For example:

```
samtools view -b NA12812.bam \
  | excord \
  --discordantdistance 500 \
  --fasta hs37d5.fa.gz \
  /dev/stdin \
  | LC_ALL=C sort−buffer-size 2 G -k1,1 -k2,2n -k3,3n \
  | bgzip -c > alt/NA12812.bed.gz
```

Once all sample BED files have been processed an index is created using giggle. For example:

```
giggle index -i "alt/*gz" -o alt_idx -s -f
```

The last step is to create a sample database from a cohort pedigree file (PED). At a minimum, this file must contain a file header, and one line per sample where each line must contain the sample name and the name of its associated BED file. The following example has three extra fields:

| Sample | Sex | Population | Super_Population | Alt_File |
|--------|-----|------------|------------------|----------|
| NA12812 | 1 | CEU | EUR | NA12812.bed.gz |
| HG00672 | 2 | CHS | EAS | HG00672.bed.gz |
| NA12878 | 2 | CEU | EUR | NA12878.bed.gz |
| HG00674 | 1 | CHS | EAS | HG00674.bed.gz |

Creating the sample database requires the giggle index, input PED file name, output database name and the column number that contains the name of the sample BED file. For example:

```
stix -i alt_idx -p four.ped -d four.ped.db -c 5
```

Once the BED files have been indexed and the sample database has been created from the PED file, STIX can now query the samples for SV evidence. For each query, the user must specify the index location (-i), sample database (-d), SV type (-t),left (-l) and right (-r) breakpoint coordinates and window size (-s) to consider around each breakpoint. The window size will depend on the size and variance of the sample fragments. We recommend using the same value used for the discordant distance parameter in the excord extraction. The output of STIX is a per-sample count of alignments that support the existence of the SV in the sample. For example:

```
stix \
  -i alt_idz \
  -d four.ped.db \
  -s 500 \
  -t DEL \
  -l 14:68603030-68603035 \
  -r 14:68603738-68603743
```

| Id | Sample | Sex | Population | Super_Population | Alt_File | Pairend | Split |
|----|--------|-----|------------|------------------|----------|---------|-------|
| 0 | HG00672 | 2 | CHS | EAS | HG00672.13.14.bed.gz | 8 | 0 |
| 1 | HG00674 | 1 | CHS | EAS | HG00674.13.14.bed.gz | 7 | 0 |
| 2 | NA12812 | 1 | CEU | EUR | NA12812.13.14.bed.gz | 7 | 0 |
| 3 | NA12878 | 2 | CEU | EUR | NA12878.13.14.bed.gz | 11 | 0 |

**1,000 genomes phase three STIX index.** For this, 2,504 low-coverage BAMs (GRCh37) and the PED file were downloaded from the 1,000 genomes AWS S3 bucket (s3://1000genomes/phase3/data/). Excord was run on each sample with --discordantdistance set to 500.

**Simons Genome Diversity Panel STIX index.** For this, 252 30× coverage FASTQ files and PED file were downloaded from the Simons Foundation (https://www.simonsfoundation.org/simons-genome-diversity-project/) and aligned to the human reference genome (GRCh37) using BWA-MEM. Excord was run on each sample with --discordantdistance set to 500.

**STIX speed measurement.** To test the speed of STIX versus any other alternative genotyping method that accesses the BAMs directly, we compared the time required for STIX to query a specific SV (DEL, 10:105053143-105054173) across the full 1KG cohort versus how much time was required to read the alignments in the same region of each BAM in the 1KG cohort. The assumption being that any genotyping method would need to at least read the alignments, and the I/O time would be a lower bound for any such method.

```
$ time stix \
  -i 1kg_stix_idx \
  -d 1kg.ped.db \
  -s 500 \
  -t DEL \
  -l 10:105053143-105053143 \
  -r 10:105054173-105054173 -S>/dev/null
real 0m1.531s
$ time ls 1000G_phaseIII_whole_genome/*.mapped.*.low_coverage.*.bam \
  | gargs 'samtools view {} 10:105052643-105053143>/dev/null'
real 16m45.827s
```

**Source code availability and Snakemake pipeline.** To improve readability and reproducatiblity, the source code for all experiments and analysis in this paper are part of a Snakemake[20] pipeline available at https://github.com/ryanlayer/stix_paper/blob/main/Snakefile. In the following sections, the relevant rules within the pipeline are listed.

**Accuracy measurement.** To determine STIX's classification performance, we considered the 1KG cohort and the phase three SVs identified by Sudmant et al.[7]. For each reported deletion, duplication and inversion, we collected the samples that were identified by 1KG as being nonreference. This analysis only included SVs with the CIEND and CIPOS VCF info field values specified.

For each of those SVs, we then constructed a similar list of samples where STIX found evidence of the same variant.

Given the list of nonreference samples from the 1KG catalog and the list of samples with supporting evidence from STIX, we computed the following values for deletions, duplications and inversions separately.

- positives (P): number of nonreference samples in the 1KG catalog
- negatives (N): number of reference samples in the 1KG catalog (total samples minus positives)
- true positives (TP): number of samples with evidence from STIX that were nonreference in the 1KG catalog
- true negatives (TN): number of samples with no evidence from STIX that were reference in the 1KG catalog
- false positives (FP): number of samples with evidence from STIX that were reference in the 1KG catalog
- false negatives (FN): number of samples with no evidence from STIX that were nonreference in the 1KG catalog
  From these values we computed:
- accuracy $= (TP + TN)/(P + N)$
- precision $= TP/(TP + FP)$
- sensitivity $= TP/P$
- specificity $= TN/N$
- F1 $= 2TP/(2TP + FP + FN)$

*Relevant Snakemake rules.*

- onekg_classification_stats
- onekg_sv_table

**COSMIC SV evaluation.** The COSMIC SV catalog was downloaded from the COSMIC website (https://cancer.sanger.ac.uk/cosmic/download, Structural Genomic Rearrangements, login required). The chromosomal position of the deletions (intrachromosomal deletion), duplications (intrachromosomal tandem duplication) and inversions (intrachromosomal inversion) were extracted and sorted into a compressed BED file.

*Relevant Snakemake rules:.*

- cosmic_sv_beds
  To determine the overlap between the COSMIC SVs and the 1KG catalog, we converted the 1KG SV VCF to SV-type-specific BED files and intersected these files with the corresponding COSMIC BED files. Intersections required a reciprocal overlap of 90%. From these intersections, we compute the 1KG allele frequency.
- onekg_gts
- Cosmic_1kg_overlap

*Relevant scripts:.*

- src/get_1kg_ac.py
  To determine the overlap between the COSMIC SVs and the gnomAD SV catalog, we retrieved the v.2.1 SV BED file from the gnomAD website (https://gnomad.broadinstitute.org/downloads/#v2-structural-variants) and split the BED file into SV-type-specific BED files and intersected these files with the corresponding COSMIC BED files. Intersections required a reciprocal overlap of 90%.

*Relevant Snakemake rules:.*

- cosmic_gnomad_overlap
  To determine the overlap between the COSMIC SVs and the STIX for 1KG and SGDP, we submitted a STIX query for each SV in the COSMIC SV-type BED files using a 500 base pair window. For each SV, we compute the number of samples with some supporting evidence.

*Relevant Snakemake rules:.*

- cosmic_stix_1kg_overlap_stats

*Relevant scripts:.*

- src/qdel.sh

**PCAWG SV evaluation.**
- The PCAWG SV catalogs were downloaded from the International Cancer Genome Consortium (ICGC) data portal website (https://dcc.icgc.org/releases/PCAWG/consensus_sv/) and combined SV-type-specific call sets.

*Relevant scripts:.*

- src/get_pcawg_svs.sh
  Similar to the process in COSMIC SV evaluation, to determine the overlap between the PCAWG SVs and the 1KG catalog, we converted the 1KG SV VCF to SV-type-specific BED files and intersected these files with the corresponding PCAWG BED files. Intersections required a reciprocal overlap of 90%. From these intersections we compute the 1KG allele frequency.

*Relevant Snakemake rules:.*

- pcawg_1kg_overlap
  To determine the overlap between the PCAWG SVs and the gnomAD SV catalog, we retrieved the v.2.1 SV BED file from the gnomAD website (https://gnomad.broadinstitute.org/downloads/#v2-structural-variants) and split the BED file into SV-type-specific BED files and intersected these files

with the corresponding PCAWG BED files. Intersections required a reciprocal overlap of 90%.

*Relevant Snakemake rules:.*

- pcawg_gnomad_overlap
  To determine the overlap between the PCAWG SVs and the STIX for 1KG and SGDP, we submitted a STIX query for each SV in the PCAWG SV-type BEDPE files using a 500 base pair window. For each SV, we compute the number of samples with some supporting evidence.

*Relevant scripts:.*

- src/get_pcawg_stix_1kg_overlap.sh
- src/get_pcawg_stix_sgdp_overlap.sh

**De novo SV evaluation.**
- The de novo SV catalog was retrieved from the GitHub repository referenced in the publication. Those SVs were reported using the GRCh38 human reference genome. We used the University of California, Santa Cruz genome browser tools to lift the SVs to GRCH37, then split the file into SV-type-specific BED files.

*Relevant Snakemake rules:.*

- denovo_sv_beds
  Similar to the process in COSMIC SV evaluation, to determine the overlap between the de novo SVs and the 1KG catalog, we converted the 1KG SV VCF to SV-type-specific BED files and intersected these files with the corresponding PCAWG BED files. Intersections required a reciprocal overlap of 90%. From these intersections, we compute the 1KG allele frequency.

*Relevant Snakemake rules:.*

- denovo_1kg_overlap
  To determine the overlap between the de novo SVs and the gnomAD SV catalog, we retrieved the v.2.1 SV BED file from the gnomAD website (https://gnomad.broadinstitute.org/downloads/#v2-structural-variants) and split the BED file into SV-type-specific BED files and intersected these files with the corresponding PCAWG BED files. Intersections required a reciprocal overlap of 90%.

*Relevant Snakemake rules:.*

- denovo_gnomad_overlap
  To determine the overlap between the de novo SVs and the STIX for 1KG and SGDP, we submitted a STIX query for each SV in the de novo SV-type BED files using a 500 base pair window. For each SV we compute the number of samples with some supporting evidence.

*Relevant Snakemake rules:.*

- denovo_stix_1kg_overlap
- denovo_stix _sgdp_overlap

**STIX germline filtering evaluation.** Information regarding PCAWG donor IDs, file IDs and specimen type can be found in Supplementary Table 4. Additionally, we have provided a table mapping PCAWG file ID to BAM sample name for BAMs used for MANTA SV calls in Supplementary Table 6. We used MANTA v.1.6.0 to call SVs in the PCAWG samples. For each tumor, we called SVs in normal mode as well as matched tumor/normal mode.

To create MANTA SV calling workflows for the ICGC samples, we used the following commands:

*Single sample (normal) mode.* $MANTA_INSTALL_PATH/bin/configManta.py \
    --bam $BAM_PATH \
    --referenceFasta $REF_GENOME_PATH \
    --runDir $OUTPUT_DIRECTORY

*Paired tumor-normal mode.* $MANTA_INSTALL_PATH/bin/configManta.py \
    --normalBam $NORMAL_BAM \
    --tumorBAM $TUMOR_BAM \
    --referenceFasta $REF_GENOME_PATH \
    --runDir $OUTPUT_DIRECTORY

All BAMs are aligned to the hs37d5 reference genome, which can be downloaded via the 1KG ftp (ftp://ftp.1000genomes.ebi.ac.uk/vol1/ftp/technical/reference/phase2_reference_assembly_sequence/hs37d5.fa.gz). After running the MANTA configuration script, a runWorkflow.py script is generated in the designated run directory and can be run as follows: ./runWorkflow.py -j $THREADS

The germline filtering analysis pipeline and associated scripts are contained within a Snakemake pipeline located at https://github.com/ryanlayer/stix_paper/tree/main/germline_filtering/stix.smk. Instructions for how to install dependencies and run the pipeline can be found under germline_filtering/README.md.

The pipeline performs the 1KG STIX queries using the deletion SVs called from the MANTA normal mode call sets. Regions that return evidence from the 1KG STIX query are filtered out. For comparison, we then perform filtering by subtracting sets of deletion regions in GnomAD and 1KG, respectively. For evaluation we intersect the STIX, GnomAD and 1KG filtered regions along with the MANTA tumor/normal SV calls with the PCAWG somatic deletion SVs for each sample. All intersection and subtraction operations were performed with a 90% reciprocal overlap threshold. False positives were the SVs that passed the filters but were not in the PCAWG calls. True positives were the SVs that passed the filters and were in the PCAWG calls. False negatives were SVs that did not pass the filters and were in the PCAWG calls.

**STIX query resolution evaluation.** For the 31,762 deletions in the 1KG call set that STIX also found evidence for, we shifted the start and end coordinates up and downstream 500 bp at 50-bp steps. At each step we submitted the STIX query with the new coordinates and counted the number of samples with supporting evidence, then computed the proportion of the number of samples at each step to the number of samples found by the original query. Finally, we plotted the median of proportions at each step.

*Relevant Snakemake rules:.*

- stix_1kg_deletion_resolution_slide
- Stix_1kg_deletion_resolution_plot

**Reporting Summary.** Further information on research design is available in the Nature Research Reporting Summary linked to this article.

## Data availability
For most data availability, the Snakemake pipeline provided by https://github.com/ryanlayer/stix_paper downloads data used for analyses. For the somatic SV filtering analysis done using PCAWG alignment files, access to data is restricted. Information regarding PCAWG sample data used for this analysis can be found under the Methods subsection STIX germline filtering evaluation.

## Code availability
STIX source code can be found at https://github.com/ryanlayer/stix. Excord source code can be found at https://github.com/brentp/excord. Source code for data analysis can be found at https://github.com/ryanlayer/stix_paper

## References

20. Köster, J. & Rahmann, S. Snakemake—a scalable bioinformatics workflow engine. *Bioinformatics* **28**, 2520–2522 (2012).

## Acknowledgements
R.M.L. was funded by NIH/NHGRI grant no. R00 HG009532 and NIH/NCI grant no. UO1 CA231978. A.R.Q. was funded by NIH/NHGRI grant no. R01 HG010757.

## Author contributions

M.C. developed software and designed, executed and analyzed experiments. B.S.P. conceived the study and developed software. F.J.S. conceived the study. A.R.Q. conceived the study. R.M.L. conceived the study, developed software and designed, executed and analyzed experiments. All authors wrote and edited the manuscript.

## Competing interests
The authors declare no competing interests.

## Additional information

**Correspondence and requests for materials** should be addressed to Ryan M. Layer.

# Reporting Summary

Please do not complete any field with "not applicable" or n/a. Refer to the help text for what text to use if an item is not relevant to your study.
For final submission: please carefully check your responses for accuracy; you will not be able to make changes later.

## Statistics

For all statistical analyses, confirm that the following items are present in the figure legend, table legend, main text, or Methods section.

| n/a | Confirmed | |
|---|---|---|
| ☐ | ⊠ | The exact sample size (n) for each experimental group/condition, given as a discrete number and unit of measurement |
| ⊠ | ☐ | A statement on whether measurements were taken from distinct samples or whether the same sample was measured repeatedly |
| ⊠ | ☐ | The statistical test(s) used AND whether they are one- or two-sided *Only common tests should be described solely by name; describe more complex techniques in the Methods section.* |
| ⊠ | ☐ | A description of all covariates tested |
| ⊠ | ☐ | A description of any assumptions or corrections, such as tests of normality and adjustment for multiple comparisons |
| ⊠ | ☐ | A full description of the statistical parameters including central tendency (e.g. means) or other basic estimates (e.g. regression coefficient) AND variation (e.g. standard deviation) or associated estimates of uncertainty (e.g. confidence intervals) |
| ⊠ | ☐ | For null hypothesis testing, the test statistic (e.g. $F$, $t$, $r$) with confidence intervals, effect sizes, degrees of freedom and $P$ value noted *Give P values as exact values whenever suitable.* |
| ⊠ | ☐ | For Bayesian analysis, information on the choice of priors and Markov chain Monte Carlo settings |
| ⊠ | ☐ | For hierarchical and complex designs, identification of the appropriate level for tests and full reporting of outcomes |
| ⊠ | ☐ | Estimates of effect sizes (e.g. Cohen's $d$, Pearson's $r$), indicating how they were calculated |

*Our web collection on statistics for biologists contains articles on many of the points above.*

## Software and code

Policy information about availability of computer code

| Data collection | Samtools >=1.10, bcftools >=1.10, giggle, STIX, standard BASH commands -- Dependencies are managed by conda >= 4.10.3 and Snakemake 6.0.6 |
|---|---|
| Data analysis | Samtools >=1.10, bcftools >=1.10, giggle, STIX, standard BASH commands -- Dependencies are managed by conda >=4.10.3 and Snakemake 6.0.6 |

For manuscripts utilizing custom algorithms or software that are central to the research but not yet described in published literature, software must be made available to editors and reviewers. We strongly encourage code deposition in a community repository (e.g. GitHub). See the Nature Portfolio guidelines for submitting code & software for further information.

## Data

Policy information about availability of data

All manuscripts must include a data availability statement. This statement should provide the following information, where applicable:
- Accession codes, unique identifiers, or web links for publicly available datasets
- A description of any restrictions on data availability
- For clinical datasets or third party data, please ensure that the statement adheres to our policy

All data analysis steps (including automatic data acquisition) are made available at https://github.com/ryanlayer/stix_paper. For protected data from PCAWG, information regarding data can be found in the data availability statement , methods, and supplementary information.

# Field-specific reporting

Please select the one below that is the best fit for your research. If you are not sure, read the appropriate sections before making your selection.

☒ Life sciences ☐ Behavioural & social sciences ☐ Ecological, evolutionary & environmental sciences

For a reference copy of the document with all sections, see nature.com/documents/nr-reporting-summary-flat.pdf

# Life sciences study design

All studies must disclose on these points even when the disclosure is negative.

| | |
|---|---|
| Sample size | 1000 Genomes (1KG) had 2504 samples, Simons Genomes Diversity Project (SGDP) had 279 samples, we used 347 PCAWG prostate cancer alignment files. For 1KG and SGDP this sample size is simply the number of samples available in the public data repositories. For PCAWG, the sample size is the number of samples that were available in the AWS cloud storage environment. |
| Data exclusions | no exclusions |
| Replication | No replication.  This studies' purpose was to demonstrate STIX and its applications with various whole genome sequencing data repositories. |
| Randomization | No randomization.  STIX is a deterministic method, and all analysis was done with preexisting data sources. |
| Blinding | No blinding.  All analysis was done using preexisting data from other sources. |

# Behavioural & social sciences study design

All studies must disclose on these points even when the disclosure is negative.

| | |
|---|---|
| Study description | Briefly describe the study type including whether data are quantitative, qualitative, or mixed-methods (e.g. qualitative cross-sectional, quantitative experimental, mixed-methods case study). |
| Research sample | State the research sample (e.g. Harvard university undergraduates, villagers in rural India) and provide relevant demographic information (e.g. age, sex) and indicate whether the sample is representative. Provide a rationale for the study sample chosen. For studies involving existing datasets, please describe the dataset and source. |
| Sampling strategy | Describe the sampling procedure (e.g. random, snowball, stratified, convenience). Describe the statistical methods that were used to predetermine sample size OR if no sample-size calculation was performed, describe how sample sizes were chosen and provide a rationale for why these sample sizes are sufficient. For qualitative data, please indicate whether data saturation was considered, and what criteria were used to decide that no further sampling was needed. |
| Data collection | Provide details about the data collection procedure, including the instruments or devices used to record the data (e.g. pen and paper, computer, eye tracker, video or audio equipment) whether anyone was present besides the participant(s) and the researcher, and whether the researcher was blind to experimental condition and/or the study hypothesis during data collection. |
| Timing | Indicate the start and stop dates of data collection. If there is a gap between collection periods, state the dates for each sample cohort. |
| Data exclusions | If no data were excluded from the analyses, state so OR if data were excluded, provide the exact number of exclusions and the rationale behind them, indicating whether exclusion criteria were pre-established. |
| Non-participation | State how many participants dropped out/declined participation and the reason(s) given OR provide response rate OR state that no participants dropped out/declined participation. |
| Randomization | If participants were not allocated into experimental groups, state so OR describe how participants were allocated to groups, and if allocation was not random, describe how covariates were controlled. |

# Ecological, evolutionary & environmental sciences study design

All studies must disclose on these points even when the disclosure is negative.

| | |
|---|---|
| Study description | Briefly describe the study. For quantitative data include treatment factors and interactions, design structure (e.g. factorial, nested, hierarchical), nature and number of experimental units and replicates. |
| Research sample | Describe the research sample (e.g. a group of tagged Passer domesticus, all Stenocereus thurberi within Organ Pipe Cactus National Monument), and provide a rationale for the sample choice. When relevant, describe the organism taxa, source, sex, age range and |

*any manipulations. State what population the sample is meant to represent when applicable. For studies involving existing datasets, describe the data and its source.*

| | |
|---|---|
| Sampling strategy | *Note the sampling procedure. Describe the statistical methods that were used to predetermine sample size OR if no sample-size calculation was performed, describe how sample sizes were chosen and provide a rationale for why these sample sizes are sufficient.* |
| Data collection | *Describe the data collection procedure, including who recorded the data and how.* |
| Timing and spatial scale | *Indicate the start and stop dates of data collection, noting the frequency and periodicity of sampling and providing a rationale for these choices. If there is a gap between collection periods, state the dates for each sample cohort. Specify the spatial scale from which the data are taken* |
| Data exclusions | *If no data were excluded from the analyses, state so OR if data were excluded, describe the exclusions and the rationale behind them, indicating whether exclusion criteria were pre-established.* |
| Reproducibility | *Describe the measures taken to verify the reproducibility of experimental findings. For each experiment, note whether any attempts to repeat the experiment failed OR state that all attempts to repeat the experiment were successful.* |
| Randomization | *Describe how samples/organisms/participants were allocated into groups. If allocation was not random, describe how covariates were controlled. If this is not relevant to your study, explain why.* |
| Blinding | *Describe the extent of blinding used during data acquisition and analysis. If blinding was not possible, describe why OR explain why blinding was not relevant to your study.* |

Did the study involve field work?  ☐ Yes  ☐ No

## Field work, collection and transport

| | |
|---|---|
| Field conditions | *Describe the study conditions for field work, providing relevant parameters (e.g. temperature, rainfall).* |
| Location | *State the location of the sampling or experiment, providing relevant parameters (e.g. latitude and longitude, elevation, water depth).* |
| Access & import/export | *Describe the efforts you have made to access habitats and to collect and import/export your samples in a responsible manner and in compliance with local, national and international laws, noting any permits that were obtained (give the name of the issuing authority, the date of issue, and any identifying information).* |
| Disturbance | *Describe any disturbance caused by the study and how it was minimized.* |

# Reporting for specific materials, systems and methods

We require information from authors about some types of materials, experimental systems and methods used in many studies. Here, indicate whether each material, system or method listed is relevant to your study. If you are not sure if a list item applies to your research, read the appropriate section before selecting a response.

### Materials & experimental systems

| n/a | Involved in the study |
|---|---|
| ☒ | ☐ Antibodies |
| ☒ | ☐ Eukaryotic cell lines |
| ☒ | ☐ Palaeontology and archaeology |
| ☒ | ☐ Animals and other organisms |
| ☒ | ☐ Human research participants |
| ☒ | ☐ Clinical data |
| ☒ | ☐ Dual use research of concern |

### Methods

| n/a | Involved in the study |
|---|---|
| ☒ | ☐ ChIP-seq |
| ☒ | ☐ Flow cytometry |
| ☒ | ☐ MRI-based neuroimaging |

## Antibodies

| | |
|---|---|
| Antibodies used | *Describe all antibodies used in the study; as applicable, provide supplier name, catalog number, clone name, and lot number.* |
| Validation | *Describe the validation of each primary antibody for the species and application, noting any validation statements on the manufacturer's website, relevant citations, antibody profiles in online databases, or data provided in the manuscript.* |

## Eukaryotic cell lines

Policy information about cell lines

| | |
|---|---|
| Cell line source(s) | *State the source of each cell line used.* |

| Authentication | *Describe the authentication procedures for each cell line used OR declare that none of the cell lines used were authenticated.* |
|---|---|
| Mycoplasma contamination | *Confirm that all cell lines tested negative for mycoplasma contamination OR describe the results of the testing for mycoplasma contamination OR declare that the cell lines were not tested for mycoplasma contamination.* |
| Commonly misidentified lines (See ICLAC register) | *Name any commonly misidentified cell lines used in the study and provide a rationale for their use.* |

## Palaeontology and Archaeology

| Specimen provenance | *Provide provenance information for specimens and describe permits that were obtained for the work (including the name of the issuing authority, the date of issue, and any identifying information). Permits should encompass collection and, where applicable, export.* |
|---|---|
| Specimen deposition | *Indicate where the specimens have been deposited to permit free access by other researchers.* |
| Dating methods | *If new dates are provided, describe how they were obtained (e.g. collection, storage, sample pretreatment and measurement), where they were obtained (i.e. lab name), the calibration program and the protocol for quality assurance OR state that no new dates are provided.* |

☐ Tick this box to confirm that the raw and calibrated dates are available in the paper or in Supplementary Information.

| Ethics oversight | *Identify the organization(s) that approved or provided guidance on the study protocol, OR state that no ethical approval or guidance was required and explain why not.* |
|---|---|

Note that full information on the approval of the study protocol must also be provided in the manuscript.

## Animals and other organisms

Policy information about studies involving animals; ARRIVE guidelines recommended for reporting animal research

| Laboratory animals | *For laboratory animals, report species, strain, sex and age OR state that the study did not involve laboratory animals.* |
|---|---|
| Wild animals | *Provide details on animals observed in or captured in the field; report species, sex and age where possible. Describe how animals were caught and transported and what happened to captive animals after the study (if killed, explain why and describe method; if released, say where and when) OR state that the study did not involve wild animals.* |
| Field-collected samples | *For laboratory work with field-collected samples, describe all relevant parameters such as housing, maintenance, temperature, photoperiod and end-of-experiment protocol OR state that the study did not involve samples collected from the field.* |
| Ethics oversight | *Identify the organization(s) that approved or provided guidance on the study protocol, OR state that no ethical approval or guidance was required and explain why not.* |

Note that full information on the approval of the study protocol must also be provided in the manuscript.

## Human research participants

Policy information about studies involving human research participants

| Population characteristics | *Describe the covariate-relevant population characteristics of the human research participants (e.g. age, gender, genotypic information, past and current diagnosis and treatment categories). If you filled out the behavioural & social sciences study design questions and have nothing to add here, write "See above."* |
|---|---|
| Recruitment | *Describe how participants were recruited. Outline any potential self-selection bias or other biases that may be present and how these are likely to impact results.* |
| Ethics oversight | *Identify the organization(s) that approved the study protocol.* |

Note that full information on the approval of the study protocol must also be provided in the manuscript.

## Clinical data

Policy information about clinical studies

All manuscripts should comply with the ICMJE guidelines for publication of clinical research and a completed CONSORT checklist must be included with all submissions.

| Clinical trial registration | *Provide the trial registration number from ClinicalTrials.gov or an equivalent agency.* |
|---|---|
| Study protocol | *Note where the full trial protocol can be accessed OR if not available, explain why.* |
| Data collection | *Describe the settings and locales of data collection, noting the time periods of recruitment and data collection.* |
| Outcomes | *Describe how you pre-defined primary and secondary outcome measures and how you assessed these measures.* |

# Dual use research of concern

Policy information about dual use research of concern

## Hazards

Could the accidental, deliberate or reckless misuse of agents or technologies generated in the work, or the application of information presented in the manuscript, pose a threat to:

| No | Yes | |
|----|-----|--|
| ☒ | ☐ | Public health |
| ☒ | ☐ | National security |
| ☒ | ☐ | Crops and/or livestock |
| ☒ | ☐ | Ecosystems |
| ☒ | ☐ | Any other significant area |

## Experiments of concern

Does the work involve any of these experiments of concern:

| No | Yes | |
|----|-----|--|
| ☒ | ☐ | Demonstrate how to render a vaccine ineffective |
| ☒ | ☐ | Confer resistance to therapeutically useful antibiotics or antiviral agents |
| ☒ | ☐ | Enhance the virulence of a pathogen or render a nonpathogen virulent |
| ☒ | ☐ | Increase transmissibility of a pathogen |
| ☒ | ☐ | Alter the host range of a pathogen |
| ☒ | ☐ | Enable evasion of diagnostic/detection modalities |
| ☒ | ☐ | Enable the weaponization of a biological agent or toxin |
| ☒ | ☐ | Any other potentially harmful combination of experiments and agents |

# ChIP-seq

## Data deposition

☐ Confirm that both raw and final processed data have been deposited in a public database such as GEO.

☐ Confirm that you have deposited or provided access to graph files (e.g. BED files) for the called peaks.

| | |
|--|--|
| Data access links *May remain private before publication.* | *For "Initial submission" or "Revised version" documents, provide reviewer access links. For your "Final submission" document, provide a link to the deposited data.* |
| Files in database submission | *Provide a list of all files available in the database submission.* |
| Genome browser session (e.g. UCSC) | *Provide a link to an anonymized genome browser session for "Initial submission" and "Revised version" documents only, to enable peer review. Write "no longer applicable" for "Final submission" documents.* |

## Methodology

| | |
|--|--|
| Replicates | *Describe the experimental replicates, specifying number, type and replicate agreement.* |
| Sequencing depth | *Describe the sequencing depth for each experiment, providing the total number of reads, uniquely mapped reads, length of reads and whether they were paired- or single-end.* |
| Antibodies | *Describe the antibodies used for the ChIP-seq experiments; as applicable, provide supplier name, catalog number, clone name, and lot number.* |
| Peak calling parameters | *Specify the command line program and parameters used for read mapping and peak calling, including the ChIP, control and index files used.* |
| Data quality | *Describe the methods used to ensure data quality in full detail, including how many peaks are at FDR 5% and above 5-fold enrichment.* |
| Software | *Describe the software used to collect and analyze the ChIP-seq data. For custom code that has been deposited into a community repository, provide accession details.* |

# Flow Cytometry

## Plots

Confirm that:

☐ The axis labels state the marker and fluorochrome used (e.g. CD4-FITC).

☐ The axis scales are clearly visible. Include numbers along axes only for bottom left plot of group (a 'group' is an analysis of identical markers).

☐ All plots are contour plots with outliers or pseudocolor plots.

☐ A numerical value for number of cells or percentage (with statistics) is provided.

## Methodology

| | |
|---|---|
| Sample preparation | *Describe the sample preparation, detailing the biological source of the cells and any tissue processing steps used.* |
| Instrument | *Identify the instrument used for data collection, specifying make and model number.* |
| Software | *Describe the software used to collect and analyze the flow cytometry data. For custom code that has been deposited into a community repository, provide accession details.* |
| Cell population abundance | *Describe the abundance of the relevant cell populations within post-sort fractions, providing details on the purity of the samples and how it was determined.* |
| Gating strategy | *Describe the gating strategy used for all relevant experiments, specifying the preliminary FSC/SSC gates of the starting cell population, indicating where boundaries between "positive" and "negative" staining cell populations are defined.* |

☐ Tick this box to confirm that a figure exemplifying the gating strategy is provided in the Supplementary Information.

# Magnetic resonance imaging

## Experimental design

| | |
|---|---|
| Design type | *Indicate task or resting state; event-related or block design.* |
| Design specifications | *Specify the number of blocks, trials or experimental units per session and/or subject, and specify the length of each trial or block (if trials are blocked) and interval between trials.* |
| Behavioral performance measures | *State number and/or type of variables recorded (e.g. correct button press, response time) and what statistics were used to establish that the subjects were performing the task as expected (e.g. mean, range, and/or standard deviation across subjects).* |

## Acquisition

| | |
|---|---|
| Imaging type(s) | *Specify: functional, structural, diffusion, perfusion.* |
| Field strength | *Specify in Tesla* |
| Sequence & imaging parameters | *Specify the pulse sequence type (gradient echo, spin echo, etc.), imaging type (EPI, spiral, etc.), field of view, matrix size, slice thickness, orientation and TE/TR/flip angle.* |
| Area of acquisition | *State whether a whole brain scan was used OR define the area of acquisition, describing how the region was determined.* |

Diffusion MRI    ☐ Used    ☐ Not used

## Preprocessing

| | |
|---|---|
| Preprocessing software | *Provide detail on software version and revision number and on specific parameters (model/functions, brain extraction, segmentation, smoothing kernel size, etc.).* |
| Normalization | *If data were normalized/standardized, describe the approach(es): specify linear or non-linear and define image types used for transformation OR indicate that data were not normalized and explain rationale for lack of normalization.* |
| Normalization template | *Describe the template used for normalization/transformation, specifying subject space or group standardized space (e.g. original Talairach, MNI305, ICBM152) OR indicate that the data were not normalized.* |
| Noise and artifact removal | *Describe your procedure(s) for artifact and structured noise removal, specifying motion parameters, tissue signals and physiological signals (heart rate, respiration).* |

Volume censoring

*Define your software and/or method and criteria for volume censoring, and state the extent of such censoring.*

## Statistical modeling & inference

Model type and settings

*Specify type (mass univariate, multivariate, RSA, predictive, etc.) and describe essential details of the model at the first and second levels (e.g. fixed, random or mixed effects; drift or auto-correlation).*

Effect(s) tested

*Define precise effect in terms of the task or stimulus conditions instead of psychological concepts and indicate whether ANOVA or factorial designs were used.*

Specify type of analysis:  ☐ Whole brain    ☐ ROI-based    ☐ Both

Statistic type for inference
(See Eklund et al. 2016)

*Specify voxel-wise or cluster-wise and report all relevant parameters for cluster-wise methods.*

Correction

*Describe the type of correction and how it is obtained for multiple comparisons (e.g. FWE, FDR, permutation or Monte Carlo).*

## Models & analysis

n/a | Involved in the study
☐ | ☐ Functional and/or effective connectivity
☐ | ☐ Graph analysis
☐ | ☐ Multivariate modeling or predictive analysis

Functional and/or effective connectivity

*Report the measures of dependence used and the model details (e.g. Pearson correlation, partial correlation, mutual information).*

Graph analysis

*Report the dependent variable and connectivity measure, specifying weighted graph or binarized graph, subject- or group-level, and the global and/or node summaries used (e.g. clustering coefficient, efficiency, etc.).*

Multivariate modeling and predictive analysis

*Specify independent variables, features extraction and dimension reduction, model, training and evaluation metrics.*

