## [Peer Review File. · Nature Methods]

Peer Review Information

Manuscript Title: Searching Thousands of Genomes to Classify Somatic and Novel Structural Variants

Corresponding author name(s): Ryan Layer

Reviewer Comments & Decisions:

Decision Letter, initial version:

Subject: Decision on Nature Methods submission NMETH-BC45904

Message:

16th Jun 2021

Dear Dr Layer,

Your Brief Communication, "Mining Thousands of Genomes to Classify Somatic and Pathogenic Structural Variants", has now been seen by 3 reviewers. As you will see from their comments below, although the reviewers find your work of potential interest, they have raised a number of concerns. We are interested in the possibility of publishing your paper in Nature Methods, but would like to consider your response to these concerns before we reach a final decision on publication.

We therefore invite you to revise your manuscript to fully address these concerns, including performing additional analyses and improving the tool.

* include a point-by-point response to the reviewers and to any editorial suggestions

* please underline/highlight any additions to the text or areas with other significant changes to facilitate review of the revised manuscript

* address the points listed described below to conform to our open science requirements

* ensure it complies with our general format requirements as set out in our guide to authors at www.nature.com/naturemethods

* resubmit all the necessary files electronically by using the link below to access your home page

[REDACTED]

We hope to receive your revised paper within 6 weeks. We are very aware of the difficulties caused by the COVID-19 pandemic to the community. If you cannot send it within this time, please let us know. In this event, we will still be happy to reconsider your paper at a later date so long as nothing similar has been accepted for publication at Nature Methods or published elsewhere.

OPEN SCIENCE REQUIREMENTS

REPORTING SUMMARY AND EDITORIAL POLICY CHECKLISTS

Please note that these forms are dynamic ‘smart pdfs’ and must therefore be downloaded and completed in Adobe Reader. We will then flatten them for ease of use by the reviewers. If you would like to reference the guidance text as you complete the template, please access these flattened versions at <http://www.nature.com/authors/policies/availability.html>.

DATA AVAILABILITY

All novel DNA and RNA sequencing data, protein sequences, genetic polymorphisms, linked genotype and phenotype data, gene expression data, macromolecular structures, and proteomics data must be deposited in a publicly accessible database, and accession codes and associated hyperlinks must be provided in the “Data Availability” section.

Please include a “Data availability” subsection in the Online Methods. This section should inform readers about the availability of the data used to support the conclusions of your study, including accession codes to public repositories, references to source data that may be published alongside the paper, unique identifiers such as URLs to data repository entries, or data set DOIs, and any other statement about data availability. At a minimum, you should include the following statement: “The data that support the findings of this study are available from the corresponding author upon request”, describing which data is available upon request and mentioning any restrictions on availability. If DOIs are provided, please include these in the Reference list (authors, title, publisher (repository name),

identifier, year). For more guidance on how to write this section please see:

<http://www.nature.com/authors/policies/data/data-availability-statements-data-citations.pdf>

CODE AVAILABILITY

Please include a “Code Availability” subsection in the Online Methods which details how your custom code is made available. Only in rare cases (where code is not central to the main conclusions of the paper) is the statement “available upon request” allowed (and reasons should be specified).

MATERIALS AVAILABILITY

ORCID

Nature Methods is committed to improving transparency in authorship. As part of our efforts in this direction, we are now requesting that all authors identified as ‘corresponding author’ on published papers create and link their Open Researcher and Contributor Identifier (ORCID) with their account on the Manuscript Tracking System (MTS), prior to acceptance. This applies to primary research papers only. ORCID helps the scientific community achieve unambiguous attribution of all scholarly contributions. You can create and link your ORCID from the home page of the MTS by clicking on ‘Modify my Springer Nature account’. For more information please visit www.springernature.com/orcid.

Sincerely,

Lin

Lin Tang, PhD
Senior Editor
Nature Methods

Reviewers' Comments:

Reviewer #1:

Remarks to the Author:

The authors have developed an efficient algorithm and software for finding reads that support SVs in a large dataset. They have also applied this tool and provided a website. This tool is a significant advance in the field and the speed and ease of use will make SV calls easier to interpret by geneticists. The tool can be used to corroborate evidence found by other SV callers or by clinical geneticists/oncologists searching for likely causal mutations.

Comments:

Major: How exactly does STIX determine that there is evidence for a variant in another sample? This should be clearly defined in the methods and a short description given in the main text. I assume that this is somehow based on choosing parameters from the insert size distribution, but how is not clear from the text. It also could be inferred from the parameters used when running the programs, but some motivation or description should be given for why those parameters were chosen. This also speaks to the resolution of the method, the method will never be able to distinguish between two highly similar SVs, the reader should be made aware of this and some estimate of this with a statement such as “the algorithm cannot distinguish between two SVs that differ in location and length by less than XXXbp”, where XXX is some number that the authors have determined.

Some of the claims made in the paper are not supported by data, but are rather assumptions made by the authors that may or may not be correct, the claims should be adjusted accordingly.

Calling and interpreting SVs is still difficult (and will still be after the publication of this manuscript). The authors might portray this fact more clearly in the manuscript.

I can see the motivation for the assumption that somatic mutations are believed to be rare and consequently absent from SV databases. My experience however tells me that this is probably not necessarily the case. This assumption is clearly not correct for somatic SNPs. First, it is near impossible to distinguish between a somatic mutation of 50% or 100% allelic frequency and a germline mutation, making it likely that some will be present in large scale databases. Second, the human genome has highly varying fragility and recurrent SVs are not at all uncommon. Third, I suspect that similar to somatic SNPs, some SV locations are more likely to increase the fitness of the cell and consequently lead to cancer.

Technically STIX findreads that support a variant or variants that are similar to the variant. I.e. if there is strong support for a variant in STIX it does not mean that the variant is present, but rather that the variant or a similar variant is likely to be present. I appreciate that the comment may seem pedantic, but it is important for the reader to know what some of the limitations of the methods presented are. STIX is e.g. not likely to work well for causal variants within VNTRs or SVs that are near other common SVs. This e.g. means that you cannot “conclude that an SV with high level of support in healthy individuals is likely to be a germline variant or systematic noise”. Rather the method does not have power to distinguish such causal variants from germline variants or systematic noise. Similarly the SVs are not necessarily found in the 1000 genomes project, rather STIX finds evidence that supports those variants samples in the 1000 genomes project.

You have made the (reasonable) assumption that the variant you find support for in other databases are unlikely to be somatic, but you haven't quantified the likelihood. Claims similar to “The SVs found by STIX are either germline or recurrent mutations and are unlikely to be driving tumor evolution” need to be phrased more accurately.

Is there a fixed threshold for which “an SV is found by STIX”? From the main text and the website it does not appear that there is. This would greatly increase the usefulness of the program. I appreciate that determining such a threshold is difficult, if you have already done so please add the threshold to the paper and the website, if not make this clear to the reader.

Recurrent de novo SVs are well known (e.g. <https://www.nature.com/articles/nature07229>). It is not clear how common or rare such events are, claims need to be adjusted accordingly.

The short paragraph about de novo SVs is confusing to me. Are you claiming that STIX is inaccurate, that you believe that reference 21 has false positives or something else that I cannot gather from the text what is?

The manuscript refers to structural variants, but the website stix.colorado.edu only supports deletions. The evidence on the website is given as the number of reads, but it is hard to know the significance of these, reporting also the coverage and genomic average support would be helpful. Also a simple call that says "this variant/or another similar one is found in the databases" would be very much useful to the user.

The methods section might be more readable if the commands were separated from the text. A section describing the main algorithmic method should be added.

What is the motivation for using the abbreviation 1KG for the 1000 genomes project? k, but not K, can be used for kilo or 1000.

Reviewer #2:

Remarks to the Author:

Layer et al describe a tool to help leverage large WG reference population sequencing databases for SV filtering. Panels of reference normals are particularly useful for calling somatic SVs, with or without a matched normal sample. They may also be useful for identifying rare SVs for Mendelian disease analysis. The presence of even subtle read level support for a putative SV call may suggest that the variant is a polymorphism or artifact (eg of alignment).

Key challenges with using current resources for this purpose is that the published / processed SV calls from these projects have been tuned for biological discovery. While using read level data for filtering would be ideal for both somatic and rare SV analysis, read level data is very cumbersome and beyond the means of most users to download and analyze. There are also privacy / consent issues with accessing these data.

The solution proposed by the authors is a web interface STIX that builds a "GIGGLE index" of discordant reads (filtered by another tool called excord) which enables the fast pull down of SV support in the form of split and discordant counts.

Overall I think the tool addresses an important need, and could be useful in practice for both tool developers and clinical geneticists. As privacy concerns and data storage / compute issues become rate

limiting for genome analysis, solutions like those proposed here will be an important part of the algorithm and reference data ecosystem.

The paper would benefit from some applications to both justify the title, more broadly demonstrate a clear technical advance, and motivate wide adoption. Additional methodological improvements would also enhance utility.

Key concerns

- The results should better demonstrate utility of the approach. The improved ability of 1KG / SGDP STIX to detect false positives in COSMIC and PCAWG relative to the official 1KG / SGDP callsets is encouraging but possibly incremental. What qualitatively improved insight does this filtering provide? For example an improvement in the power to nominate novel frequently mutated cancer or disease genes. Or provide some orthogonal measure to show that pathogenic variants are better detected and how much so .. eg examples of near misses ie genetic misdiagnoses that STIX helped avoid.
- Matched normals are usually used for somatic SV calling. This tool could be useful in situations where a matched normal is lacking (eg cell lines). Can authors show that their tool enables detection of somatic SV without a matched normal?
- The current query format seems to take a limited vocabulary of simple SVs (DEL, INS, TRA, INV). Can the authors expand the tool to provide support for an arbitrary vcf BND or bedpe rearrangement?
- Current tool uses split reads that already exist in the BWA MEM bam. These may be an underestimate of the total read support for an SV. It should be simple for the tool to realign reads to a user-derived contig (eg obtained through the fusion of two reference sequences) at query time. The authors should either implement this or show that it doesn't make a difference.
- (related to above) Current tool only evaluates simple pairwise fusions however a user may be interested in identifying read support for more complex SVs that paste together three or more sequences or more broadly assess the support for an arbitrary contig / sequence. It would be very useful to have this functionality.
- Current output returns only counts, however an important part of visual or analytic SV evaluation is understanding more detailed aspects of alignment patterns. It would be useful if the tool returned a partial or even full alignment record including CIGAR, MD tag, MAPQ, alignment scores and/or summary stats on mapping qualities, alignment scores, and base qualities. These may be useful to both a tool developer and for visualization of variants (eg in the clinical setting).

- Can this tool lower the threshold for data access for the average user? What are some protocols to reduce or eliminate leakage or sanitize the output?

- Exord uses a fixed discordant distance but this value should be library dependent based on the insert size distribution. This could miss insertions or local deletions depending on whether this fixed value is an under or overestimate. The authors should justify this choice or make this part of the processing data driven.

Minor

- Figure 2 points are overplotted so density is unclear. In particular, not visually clear what fraction of called deletions in PCAWG and COSMIC are seen in 1KG / SGDB. At first glance it looks like most, but the Supp tables indicate otherwise.

- Captions should better describe data eg not clear from Figure 2 caption what each dot represents, presumably a deletion.

- Figure 2 caption - dataset

Reviewer #3:

Remarks to the Author:

To improve the quality of somatic and de novo SV classification, Layer et al. present STIX, a fast and sensitive method to search for SV support in large collections of samples. Using an index constructed from discordant reads in a reference panel of samples, STIX searches for evidence of the SV. This approach reduces the storage and CPU time of searching full sets of aligned reads. STIX is available as a web tool or a command-line tool. While STIX has overlap with genotypers, it is positioned as a validation tool. For example, filtering false de novo or somatic SV calls to increase confidence is difficult, and STIX fills a niche as an ultra-sensitive method. The utility of STIX is demonstrated by invalidating a set of de novo calls from 1000 Genomes and by finding evidence for somatic mutations in COSMIC and PCAWG in population samples, which I found to be compelling.

Major concerns

1) While I believe that STIX is useful, I doubt it will reach a broad audience. STIX has been available for several years, but has it had an impact on significant projects, and if so, which ones? Since most of the validation work is done (citation 19), the authors focus on de novo and somatic variants to show the

utility of STIX. It's likely STIX is faster once alignments are prepared, but the authors do not address how well other methods might already be capable of similar results.

2) When STIX reports no reads, it is impossible to know if a) there was no discordant support, b) there were no aligned reads, and c) there was an error. The issue of (b), not enough data, could be solved by simultaneously querying a read-depth BED. Whether or not this was part of STIX, but the authors could significantly improve their own results and demonstrating how to distinguish no-support from no-reads. For (c), I found both the web interface and the command-line tool would report no supporting reads even when fed garbage loci. These two issues (b and c) leave an opportunity for errors and misinterpretation of the results in other research projects using STIX.

Minor comments

1) Supplementary Table 1. Add the number of deletions, duplications, and inversions tested as a row to the table.

2) The web interface is incomplete. In 1000 genomes, "Sample_id" is missing (using Chrome). It takes a while to update, and there's no indication that it's working or finished. If two regions are queried in a row and have the same output, like no discordant support, the page doesn't change at all, so the user doesn't know if the results were updated or not. Writing the coordinates over the results table would help users to know that the results were updated.

3) The authors spend a whole paragraph rebutting pangenomes, but I think it could be summarized in a couple of sentences. I don't think pangenome genotyping is really a threat to STIX, but if you really wanted to make it point, back it up with comparisons against the vg genotyper or another pangenome approach.

4) Second paragraph, check spelling of "in chase".

5) Suggest revising "These somatic false (caused by false negatives in the control tissue) positives are wide-spread.". Parenthesis disrupt the thought.

6) By "re-emerging alleles", do you mean recurrent?

7) The manuscript relies heavily on citation 19 to address other genotypers in the field. I think the authors should a) pull a brief summary of findings from that paper (and cite it) and b) give credit to other approaches and explain what STIX does differently and why it's useful.

8) What happens if other genotypers are run with ultra-sensitive settings (if they support it)? Do you get similar results?

Author Rebuttal to Initial comments

Reviewer #1:

Remarks to the Author:

The authors have developed an efficient algorithm and software for finding reads that support SVs in a large dataset. They have also applied this tool and provided a website. This tool is a significant advance in the field and the speed and ease of use will make SV calls easier to interpret by geneticists. The tool can be used to corroborate evidence found by other SV callers or by clinical geneticists/oncologists searching for likely causal mutations.

Comments:

Major: How exactly does STIX determine that there is evidence for a variant in another sample? This should be clearly defined in the methods and a short description given in the main text. I assume that this is somehow based on choosing parameters from the insert size distribution, but how is not clear from the text. It also could be inferred from the parameters used when running the programs, but some motivation or description should be given for why those parameters were chosen. This also speaks to the resolution of the method, the method will never be able to distinguish between two highly similar SVs, the reader should be made aware of this and some estimate of this with a statement such as "the algorithm cannot distinguish between two SVs that differ in location and length by less than XXXbp", where XXX is some number that the authors have determined.

We thank the reviewer for pointing out that we did not have a complete description of the evidence identification algorithm. STIX uses the clustering algorithm that was first introduced by LUMPY to determine if the evidence in a particular sample supports an SV. For a given SV, there is a range of locations that supporting paired-end reads can reside that depend on the sample's insert size distribution and the type of SV. Consider an insert size mean of 400bp and standard deviation of 50bp, paired-end evidence for a deletion has a +/- orientation with the ends more than 500bp ($400+2*50$) up/down stream of the breakpoint. Duplications and inversions have similar configurations that are now detailed in the methods section. The current version assumes that the library statistics are same for all samples in an index, which is not always true. For future work, we will move to a model where statistics are derived from individual samples for both discordant variant extraction and queries.

The reviewer is exactly right. In some cases STIX, and all short-read SV callers, cannot determine if two pieces of evidence are associated with identical or similar SVs. In particular, when considering paired-end evidence that spans an SV, the SV's breakpoint can be in a range of locations that are distinguished. To measure this dynamic in the 1KG cohort, we reran STIX queries for 28,593 deletions called by 1KG, shifting the query location up and down stream by 50bp (up to -500bp/+500bp), and recounted the number of samples with supporting evidence (Fig. 1). There is very little change in the frequency within 200bp of the original SV coordinates. After 200bp the number of samples with the SV decreases quickly, and once the query is 400bp away from the original breakpoint no samples are recovered. We have added text to clarify this point to the discussion and added Figure 1 to the supplemental figures.

Figure 1. STIX query resolution depends on the insert size distribution of the cohort under consideration. By shifting the query coordinates of 28,593 deletions called by 1KG up and downstream in 50bp increments, and recalculating the number of samples found to still have evidence for the SV, we find the 1KG STIX queries have a resolution about about 400bp.

Some of the claims made in the paper are not supported by data, but are rather assumptions made by the authors that may or may not be correct, the claims should be adjusted accordingly.

Calling and interpreting SVs is still difficult (and will still be after the publication of this manuscript). The authors might portray this fact more clearly in the manuscript.

We completely agree and we have added text to make this point clear in the discussion.

I can see the motivation for the assumption that somatic mutations are believed to be rare and consequently absent from SV databases. My experience however tells me that this is probably not necessarily the case. This assumption is clearly not correct for somatic SNPs. First, it is near impossible to distinguish between a somatic mutation of 50% or 100% allelic frequency and a germline mutation, making it likely that some will be present in large scale databases. Second, the human genome has highly varying fragility and recurrent SVs are not at all uncommon. Third, I suspect that similar to somatic SNPs, some SV locations are more likely to increase the fitness of the cell and consequently lead to cancer.

We agree that not all somatic variants are rare. In practice, the number of variants found in a given sample is so high that we often rely on allele frequency as a prioritization technique, but it is not perfect. This is the first step in many human genetic diseases studies (e.g. mendelian, rare diseases etc.) and thus a standard that we follow and enable. We have added text to make this point clear.

Technically STIX finds reads that support a variant or variants that are similar to the variant. I.e. if there is strong support for a variant in STIX it does not mean that the variant is present, but rather that the variant or a similar variant is likely to be present. I appreciate that the comment may seem pedantic, but it is important for the reader to know what some of the limitations of the methods presented are. STIX is e.g. not likely to work well for causal variants within VNTRs or SVs that are near other common SVs.

This e.g. means that you cannot "conclude that an SV with high level of support in healthy individuals is likely to be a germline variant or systematic noise". Rather the method does not have power to distinguish such causal variants from germline variants or systematic noise. Similarly the SVs are not necessarily found in the 1000 genomes project, rather STIX finds evidence that supports those variants samples in the 1000 genomes project.

This is not a pedantic point at all. The problem of differentiating between similar SVs and two identical SVs is difficult, and with short reads sometimes impossible. STIX of course relies on the abilities of short reads to genotype variants. The point can be made that this is clearly an improvement compared to VCF comparisons. The latter is still often done by using reciprocal overlap or some other ad hoc strategy. This also impacts the generation of population/cohort level VCF files. STIX enables a more reliable comparison and thus allele frequency of the same variant or a very similar variant. We have changed our language to match what the reviewer suggests.

You have made the (reasonable) assumption that the variant you find support for in other databases are unlikely to be somatic, but you haven't quantified the likelihood. Claims similar to "The SVs found by STIX are either germline or recurrent mutations and are unlikely to be driving tumor evolution" need to be phrased more accurately.

The general assumption for many disease-causing mutations is that they are low frequency or not observable at all in a presumably healthy cohort (e.g., 1000genomes). There are many papers published that leverage this exact factor to rank variants and rely on an accurate population data base. STIX is improving the accuracy of this variant ranking/priorisation. While it is extremely unlikely that a co-observed variant (e.g., in a cancer sample and in 1000genomes) is a cancer driver it is not impossible. Thus, we changed the sentence to: "The SVs found by STIX are likely either germline or recurrent mutations and are unlikely to be driving tumor evolution"

Is there a fixed threshold for which "an SV is found by STIX"? From the main text and the website it does not appear that there is. This would greatly increase the usefulness of the program. I appreciate that determining such a threshold is difficult, if you have already done so please add the threshold to the paper and the website, if not make this clear to the reader.

The stochastic fluctuations in coverages and noise makes determining a fixed threshold difficult. We agree that digesting counts decreases the usefulness, and we are actively working on new statistics and algorithms that convert the counts that STIX current yields into more succinct description of the variant's existence and frequency. We have this topic in the discussion section.

Recurrent de novo SVs are well known (e.g. <https://www.nature.com/articles/nature07229>). It is not clear how common or rare such events are, claims need to be adjusted accordingly.

Thank you for the reference. This specific study highlights 18 CNV based on arrays that were reidentified in schizophrenia and control cases. Our observation still remains relevant and interesting, showing not only CNV but also inversions to be re-identifiable. We have adjusted the paragraph accordingly.

The short paragraph about de novo SVs is confusing to me. Are you claiming that STIX is inaccurate, that you believe that reference 21 has false positives or something else that I cannot gather from the text what is?

Given the filtering and validation done by [21], we do not think they have found false positives. We instead think that these are recurrent. We have adjusted the paragraph accordingly.

The manuscript refers to structural variants, but the website stix.colorado.edu only supports deletions. The evidence on the website is given as the number of reads, but it is hard to know the significance of these, reporting also the coverage and genomic average support would be helpful. Also a simple call that says "this variant/or another similar one is found in the databases" would be very much useful to the user.

We explored capturing the base genomic coverage of each genome in the index to provide some context to the discordant calls STIX reports. Unfortunately this required an extra 3GB per sample. For 1KG that would be an extra 7.5TB, which is nearly 70X larger than the current STIX 1KG index. While we are still exploring options that would use less space, the current disk requirement for base coverage is prohibitively expensive.

To the reviewer's earlier point, positively identifying the query SV and differentiating it with another highly similar SV is difficult and at times impossible. This problem is particularly hard when more than one variant is present in the region, which will become more likely as we consider more genomes. We are actively working on new statistics to quantify the concentration of evidence in a query region with the goals of developing a metric and visualization strategies that will help users understand the context of their query and if 1) there are other SVs in the region or 2) the region is particularly noisy and prone to false positives. We have added text to these points in the discussion.

The methods section might be more readable if the commands were separated from the text. A section describing the main algorithmic method should be added.

We have moved all of the code out of the methods section and now have pointers into the github page containing the code for all of our experiments and analysis.

What is the motivation for using the abbreviation 1KG for the 1000 genomes project? k, but not K, can be used for kilo or 1000.

Keeping all abbreviations in all capitals makes it easier for readers to identify an abbreviation.

Reviewer #2:

Remarks to the Author:

Layer et al describe a tool to help leverage large WG reference population sequencing databases for SV filtering. Panels of reference normals are particularly useful for calling somatic SVs, with or without a matched normal sample. They may also be useful for identifying rare SVs for Mendelian disease analysis. The presence of even subtle read level support for a putative SV call may suggest that the variant is a polymorphism or artifact (eg of alignment).

Key challenges with using current resources for this purpose is that the published / processed SV calls from these projects have been tuned for biological discovery. While using read level data for filtering would be ideal for both somatic and rare SV analysis, read level data is very cumbersome and beyond the means of most users to download and analyze. There are also privacy / consent issues with accessing these data.

The solution proposed by the authors is a web interface STIX that builds a "GIGGLE index" of discordant reads (filtered by another tool called excord) which enables the fast pull down of SV support in the form of split and discordant counts.

Overall I think the tool addresses an important need, and could be useful in practice for both tool developers and clinical geneticists. As privacy concerns and data storage / compute issues become rate limiting for

genome analysis, solutions like those proposed here will be an important part of the algorithm and reference data ecosystem.

The paper would benefit from some applications to both justify the title, more broadly demonstrate a clear technical advance, and motivate wide adoption. Additional methodological improvements would also enhance utility.

We want to thank this reviewer for their detailed and insightful feedback. In several cases our future work on STIX was in line with the reviewers feedback, and in others the reviewer pointed out interesting new experiments that yielded exciting results. In particular,

- 1) The results from testing STIX as a germline filter for tumor SV calls indicated that STIX is as good filtering as SV calls from matched-normal tissue. This application will be extremely useful in cases where normal tissue is not available or lacks purity.
- 2) The suggestion to allow queries with arbitrary ends motivated us to look at expanding our use case from only looking for evidence of a particular SV in the population, to looking for SVs that are potentially functionally equivalent. With an index of 181 prostate cancer samples from PCAWG, we used STIX to search for TMPRSS2-ERG fusions that matched 1) a particular SV and 2) and expanded search for any fusion that included both gene bodies. In these two searches, the number of samples with supporting evidence grew from 13 to 82.

We are confident these two applications will have broad utility.

Key concerns

- The results should better demonstrate utility of the approach. The improved ability of 1KG / SGDP STIX to detect false positives in COSMIC and PCAWG relative to the official 1KG / SGDP callsets is encouraging but possibly incremental. What qualitatively improved insight does this filtering provide? For example an improvement in the power to nominate novel frequently mutated cancer or disease genes. Or provide some orthogonal measure to show that pathogenic variants are better detected and how much so .. eg examples of near misses ie genetic misdiagnoses that STIX helped avoid.

Given the state of variant interpretation, variant analysis is a painstaking task since each variant must be considered for its role in the traits under consideration. Elane Madrids has a nice discussion of this point (Genome Medicine, 2010) whose title "The \$1,000 genome, the \$100,000 analysis?" nicely summarizes the issue. The best opportunity for reducing this burden is to reduce the number of variants that are considered. gnomAD has become the single best resource for disease analysis because the frequencies it provides can reduce the curation burden for SNVs and INDELS by an order of magnitude. The primary utility of STIX is to provide a similar improvement for SVs.

To quantify these improvements, we (motivated by the reviewer's next suggestion) used STIX to identify somatic SVs made in tumor samples. Using 181 prostate cancer samples from PCAWG, we used MANTA to call SVs in 181 tumor samples from PCAWG. We then tested four germline variant filtering strategies 1) MANTA rerun in tumor/normal mode where SVs found in the matched-normal are removed, 2) removed calls where STIX found evidence in the 1KG cohort, 3) removed calls in the 1KG SV published list, 4) removed calls found in the gnomAD SV list.

STIX removed, on average, 30% more SVs than the matched-normal strategy. This reduction is significant for two reasons. First, filtering using matched-normal tissue is already a highly optimized process and is considered the gold-standard for somatic variant identification. Any improvement to an optimized process is significant, especially when it does not require additional data collection. Second, the SVs that remain after

filtering must be further studied for their role in the disease. This process is resource intensive, and removing any germline variants directly contributes to the likelihood that the analyst can find the causative variant in their budgeted time.

- Matched normals are usually used for somatic SV calling. This tool could be useful in situations where a matched normal is lacking (eg cell lines). Can authors show that their tool enables detection of somatic SV without a matched normal?

This is a great suggestion. We added the experiment that the reviewer suggested, and we showed that STIX performed as well as match-normal tissue in removing germline SVs from tumor-sample calls. More specifically, we used MANTA to call SVs in 181 tumor samples from PCAWG. We then tested four filtering strategies 1) removed MANTA made in matched-normal samples, 2) removed calls where STIX found evidence in the 1KG cohort, 3) removed calls in the 1KG SV published list, 4) removed calls found in the gnomAD SV list. STIX removed more germline calls than the matched-normal method, while the SV lists methods kept an order of magnitude more. This point is critical because each potential germline SV is typically hand-curated. STIX and the matched-normal method retained on about 20 calls on average. The SV lists retained thousands. Of the retained SVs, STIX kept about as many true positives as the matched-normal method, but also had a higher false-negative rate. Interestingly, STIX and the SV-list methods had similar false-negative rates, meaning that the SVs that STIX removed were also called in gnomAD.

- The current query format seems to take a limited vocabulary of simple SVs (DEL, INS, TRA, INV). Can the authors expand the tool to provide support for an arbitrary vcf BND or bedpe rearrangement?

We have expanded the STIX to take arbitrary pairs of coordinates in its query. The current implementation does not fully support the BND format from the VCF spec. BNDs, according to the VCF spec, are highly complex and we are studying how we can encode the various possible configurations captured by BNDs without overwhelming the user.

- Current tool uses split reads that already exist in the BWA MEM bam. These may be a underestimate of the total read support for an SV. It should be simple for the tool to realign reads to a user-derived contig (eg obtained through the fusion of two reference sequences) at query time. The authors should either implement this or show that it doesn't make a difference.

While we think this a great idea, we believe it to be beyond the scope of this method. One of the main advantages of STIX is its fast and compact representations of SV evidence in a genome. As such, raw reads are not retained. Even if the raw reads were part of the STIX index, realigning those reads for each query would significantly increase the query times from one second to hours (or longer). We are exploring retaining unaligned reads as a means to support insertions.

- (related to above) Current tool only evaluates simple pairwise fusions however a user may be interested in identifying read support for more complex SVs that paste together three or more sequences or more broadly assess the support for an arbitrary contig / sequence. It would be very useful to have this functionality.

This is another very good idea, but implementation would be quite complicated. First, it is not clear how users would easily specify a complex event. Complex SVs from events like chromothripsis can involve dozens of breakpoints and span 10s of megabases. One possibility would be to test the density of calls in a region that interacts with one or more other loci. This interface would not require users to specify the exact order or distance between events, just the broad regions interacting. A common pattern in chromothripsis is also man interspersed deletions. This interface could also report on the extent of these events in the region under

consideration. We really like this idea, especially as we expand from germline indexes to disease, and we anticipate this will be a fruitful area of future work.

- Current output returns only counts, however an important part of visual or analytic SV evaluation is understanding more detailed aspects of alignment patterns. It would be useful if the tool returned a partial or even full alignment record including CIGAR, MD tag, MAPQ, alignment scores and/or summary stats on mapping qualities, alignment scores, and base qualities. These may be useful to both a tool developer and for visualization of variants (eg in the clinical setting).

We agree that it would be very useful to pull alignments and all of the associated metadata, but in our opinion the costs outweigh the benefit. A key advantage of STIX is that its index is orders of magnitude smaller than the associated BAM files, making it cost effective to host an index to thousands of genomes. For example, simply storing the 1KG alignments in AWS would cost \$1543.04/month, whereas the STIX 1KG index would cost \$2.41/month.

- Can this tool lower the threshold for data access for the average user? What are some protocols to reduce or eliminate leakage or sanitize the output?

The goal of STIX is to make large cohorts more accessible. The best way to make data more accessible through a web service. By reducing the disk footprint and query speed for queries that span thousands of genomes, STIX makes web-based queries (like stix.colorado.edu) possible. Part of our future work is to find statistical models that accurately represent the counts from STIX. With these models, we can eliminate leakage by only returning the models (or visualizations of the models) to down-stream users.

- Exord uses a fixed discordant distance but this value should be library dependent based on the insert size distribution. This could miss insertions or local deletions depending on whether this fixed value is an under or overestimate. The authors should justify this choice or make this part of the processing data driven.

We completely agree. While most of the samples in the 1000 Genomes Project cohort have a similar insert size distribution (from which we based our extraction and queries), there is a distinct subset where that static value overestimates the insert size and we subsequently ignore evidence for smaller deletions in those samples. Part of our future, which is now listed in the discussion, is to replace the cohort-based statistics with a collection and query process that is dynamic to an individual sample's insert size distribution.

Minor

- Figure 2 points are overplotted so density is unclear. In particular, not visually clear what fraction of called deletions in PCAWG and COSMIC are seen in 1KG / SGDB. At first glance it looks like most, but the Supplemental tables indicate otherwise.

We have updated the legends to the plots to indicate what fraction of the call sets are seen in each of the subplots.

- Captions should better describe data eg not clear from Figure 2 caption what each dot represents, presumably a deletion.

We have updated the legend to include a better description of the data.

- Figure 2 caption - dataset

Good catch. Fixed.

Reviewer #3:

Remarks to the Author:

To improve the quality of somatic and de novo SV classification, Layer et al. present STIX, a fast and sensitive method to search for SV support in large collections of samples. Using an index constructed from discordant reads in a reference panel of samples, STIX searches for evidence of the SV. This approach reduces the storage and CPU time of searching full sets of aligned reads. STIX is available as a web tool or a command-line tool. While STIX has overlap with genotypers, it is positioned as a validation tool. For example, filtering false de novo or somatic SV calls to increase confidence is difficult, and STIX fills a niche as an ultra-sensitive method. The utility of STIX is demonstrated by invalidating a set of de novo calls from 1000 Genomes and by finding evidence for somatic mutations in COSMIC and PCAWG in population samples, which I found to be compelling.

We are excited and encouraged that the reviewer felt that STIX fills a niche in the SV field and that our results were compelling.

Major concerns

1) While I believe that STIX is useful, I doubt it will reach a broad audience. STIX has been available for several years, but has it had an impact on significant projects, and if so, which ones? Since most of the validation work is done (citation 19), the authors focus on de novo and somatic variants to show the utility of STIX. It's likely STIX is faster once alignments are prepared, but the authors do not address how well other methods might already be capable of similar results.

The reviewer is right, we have been working on STIX for several years, but it has never been published and we would not expect it to have made an impact on significant projects yet. If STIX is published, we believe it will have an immediate impact on cancer and other disease analysis. The demonstration that many SVs in somatic call sets are likely common variants will likely be surprising and motivate new cancer projects to consider using STIX. The result added in this revision that STIX is an effective germline SV filter for somatic SV identification will be very interesting to any project that is trying to identify deleterious SVs but does not have matched normal tissue.

While Chandle et. al. (citation 19), which shares an author with this manuscript, did perform a comprehensive comparison of genotyper, this manuscript also included a significant validation experiment that used 47,786 SVs from the 1KG calls to determine accuracy, specificity, sensitivity and specificity (Supplemental Table 1). We have brought more of the results from Chandle et. al. into this manuscript.

STIX is intended as a resource to help disease analysis, and somatic and de novo SVs are major contributors to SV-driven disease. As such, we believe it is appropriate for our manuscript to focus on these classes.

2) When STIX reports no reads, it is impossible to know if a) there was no discordant support, b) there were no aligned reads, and c) there was an error. The issue of (b), not enough data, could be solved by simultaneously querying a read-depth BED. Whether or not this was part of STIX, but the authors could significantly improve their own results and demonstrating how to distinguish no-support from no-reads. For (c), I found both the web interface and the command-line tool would report no supporting reads even when fed garbage loci. These two issues (b and c) leave an opportunity for errors and misinterpretation of the results in other research projects using STIX.

We agree that a result of zero reads can be confusing. We explored maintaining a list of normal coverage for each genome so that users could differentiate between an absence of reads and no reads supporting the SV, but in our opinion, the costs outweigh the benefit. A key advantage of STIX is that its index is orders of magnitude smaller than the associated BAM files, making it cost-effective to host an index to thousands of genomes. For example, simply storing the 1KG alignments in AWS would cost \$1543.04/month, whereas the STIX 1KG index would cost \$2.41/month. To reduce the storage burden, we have tried rounding, binning, and compressing read counts, but with an index that includes thousands of samples, the space requirements were still too high. We could go further and collapse read counts down to copy-number estimates, but this approach would depend on the quality of those calls. Alternatively, we would identify locations in the STIX database that return no reads and record if any reads were found, which would address the specific issue raised here. This is a significant problem, and we will continue to explore solutions in future research. We have added text to the discussion about this limitation.

We are exploring an approach that collects and summarizes all of the SV evidence in the vicinity of the query that corresponds to other events. The objective of this method is to measure local noise or other events in the region, but it may offer some larger insight to users to limit misinterpretations.

Minor comments

1) Supplementary Table 1. Add the number of deletions, duplications, and inversions tested as a row to the table.

The number of tested SVs has been added.

2) The web interface is incomplete. In 1000 genomes, "Sample_id" is missing (using Chrome). It takes a while to update, and there's no indication that it's working or finished. If two regions are queried in a row and have the same output, like no discordant support, the page doesn't change at all, so the user doesn't know if the results were updated or not. Writing the coordinates over the results table would help users to know that the results were updated.

Thank you for pointing this out. The web interface is now fixed.

3) The authors spend a whole paragraph rebutting pangenomes, but I think it could be summarized in a couple of sentences. I don't think pangenome genotyping is really a threat to STIX, but if you really wanted to make it point, back it up with comparisons against the vg genotyper or another pangenome approach.

While we agree that STIX and pangenomes address different problems, we have reduced our discussion to a sentence.

4) Second paragraph, check spelling of "in chase".
Fixed. Thank you.

5) Suggest revising "These somatic false (caused by false negatives in the control tissue) positives are wide-spread.". Parenthesis disrupt the thought.

Fixed. Thank you.

6) By "re-emerging alleles", do you mean recurrent?

Yes. The revised version has the suggested wording.

7) The manuscript relies heavily on citation 19 to address other genotypers in the field. I think the authors should a) pull a brief summary of findings from that paper (and cite it) and b) give credit to other approaches and explain what STIX does differently and why it's useful.

We have brought more of the results from Chandle et. al. into this manuscript, and we have added a comparison of STIX and other methods to the supplement.

8) What happens if other genotypers are run with ultra-sensitive settings (if they support it)? Do you get similar results?

They should. SV-Typer, for example, uses a Bayesian approach to test which genotype is most likely for the allele balance at the query site. The default prior for the existence of the variant is low, and if the allele balance is low (less than 0.2) then the algorithm will identify that site as reference. An ultra-sensitive mode could be enabled by changing the prior (which would require a code change), then the existence of any alternate allele would give a non-reference genotype.

The major advance that STIX provides is speed and scalability. A query in STIX takes about one second across 2506 WGS samples. Without the index the same query would take at least 16 minutes.

Decision Letter, first revision:

Subject: Decision on Nature Methods submission NMETH-BC45904A

Message:

27th Sep 2021

Dear Dr Layer,

Your Brief Communication, "Mining Thousands of Genomes to Classify Somatic and Pathogenic Structural Variants", has now been seen by 3 reviewers. As you will see from their comments below, although the reviewers find some of their comments have been addressed, there are still important concerns that remain. We are interested in the possibility of publishing your paper in Nature Methods, but would like to consider your response to these concerns before we reach a final decision on publication.

We therefore invite you to revise your manuscript to address these concerns. Among other required revisions, the remaining points from Reviewer 2 are very important and should be well addressed with new analysis and other changes.

[REDACTED]

We hope to receive your revised paper within eight weeks. We are very aware of the difficulties caused by the COVID-19 pandemic to the community. If you cannot send it within this time, please let us know. In this event, we will still be happy to reconsider your paper at a later date so long as nothing similar has been accepted for publication at Nature Methods or published elsewhere.

OPEN SCIENCE REQUIREMENTS

REPORTING SUMMARY AND EDITORIAL POLICY CHECKLISTS

Please note that these forms are dynamic ‘smart pdfs’ and must therefore be downloaded and completed in Adobe Reader. We will then flatten them for ease of use by the reviewers. If you would like to reference the guidance text as you complete the template, please access these flattened versions at <http://www.nature.com/authors/policies/availability.html>.

DATA AVAILABILITY

All novel DNA and RNA sequencing data, protein sequences, genetic polymorphisms, linked genotype and phenotype data, gene expression data, macromolecular structures, and proteomics data must be deposited in a publicly accessible database, and accession codes and associated hyperlinks must be provided in the “Data Availability” section.

To further increase transparency, we encourage you to provide, in tabular form, the data underlying the graphical representations used in your figures. This is in addition to our data-deposition policy for specific types of experiments and large datasets. For readers, the source data will be made accessible directly from the figure legend. Spreadsheets can be submitted in .xls, .xlsx or .csv formats. Only one (1) file per figure is permitted: thus if there is a multi-paneled figure the source data for each panel should be clearly labeled in the csv/Excel file; alternately the data for a figure can be included in multiple, clearly labeled sheets in an Excel file. File sizes of up to 30 MB are permitted. When submitting source

data files with your manuscript please select the Source Data file type and use the Title field in the File Description tab to indicate which figure the source data pertains to.

Please include a “Data availability” subsection in the Online Methods. This section should inform readers about the availability of the data used to support the conclusions of your study, including accession codes to public repositories, references to source data that may be published alongside the paper, unique identifiers such as URLs to data repository entries, or data set DOIs, and any other statement about data availability. At a minimum, you should include the following statement: “The data that support the findings of this study are available from the corresponding author upon request”, describing which data is available upon request and mentioning any restrictions on availability. If DOIs are provided, please include these in the Reference list (authors, title, publisher (repository name), identifier, year). For more guidance on how to write this section please see: <http://www.nature.com/authors/policies/data/data-availability-statements-data-citations.pdf>

CODE AVAILABILITY

Please include a “Code Availability” subsection in the Online Methods which details how your custom code is made available. Only in rare cases (where code is not central to the main conclusions of the paper) is the statement “available upon request” allowed (and reasons should be specified).

For more information on our code sharing policy and requirements, please see: <https://www.nature.com/nature-research/editorial-policies/reporting-standards#availability-of-computer-code>

MATERIALS AVAILABILITY

ORCID

Sincerely,

Lin

Lin Tang, PhD
Senior Editor
Nature Methods

Reviewers' Comments:

Reviewer #1:

Remarks to the Author:

I believe that the tool provided will be useful for variant filtering. The authors have addressed most of my comments from the previous round of review and adjusted their statements appropriately.

As STIX only uses insert size distribution and not the underlying sequence its ability to distinguish between similar SVs is more limited than many other short read SV callers that use the underlying sequence information. Using sequence information inside the STIX framework would be infeasible, but the statement that all other short read SV callers have this limitation is incorrect and needs to be adjusted.

I appreciate that these are not the first authors to do so, but in a scientific communication the abbreviation K should not be used to represent kilo.

Reviewer #2:

Remarks to the Author:

Pedersen et al have revised their work STIX, which introduces a novel web based index tool to enable fast structural variant filtering using publicly available databases, including 1KG and SGDP.

The key benefit of the STIX web service is to assess read-level evidence for candidate SVs in large databases. In general, the goal of that query is to identify (and remove) SV candidates that are common polymorphisms or alignment artifacts – and thus increase the specificity of SV calling. The main use cases are (1) clinical geneticists evaluating a (rare or de novo) constitutional variant (2) cancer genomicists wanting to improve (somatic) SV identification with or without a matched normal. The key premise of the work is that variants that survive the STIX filter are more likely to be pathogenic i.e. cause cancer or constitutional disease.

During revision, they have extended STIX to enable arbitrary breakend queries and also benchmarked their analysis for "tumor only" calling of structural variants. They have not supported the main claim of the paper related to pathogenicity. Their rebuttal also suggests many of the suggested revisions to be out of the scope of the current work.

Despite the new functionality and analytic updates, the manuscript still does not convey the technical advance and utility, in my view. Some of this may be an issue with data presentation. I do think that will stand in the way of wide adoption.

Specific concerns

- The authors have not addressed the correlation of STIX with pathogenicity, which is central to the paper (part of the title) but barely examined.

Their rebuttal to this critique refers to an analysis (see below) that addresses the ability of STIX to detect somatic variants. Most somatic variants however are not pathogenic ie are not cancer drivers. Similarly, de novo variants may also not be pathogenic.

The Tmprss2-ERG analysis is included a supplementary figure but not referred to in the text (neither is the figure). It is also confusing - set up in a fundamentally different than the main application of STIX the paper. Namely, it builds a STIX index of cancer reads (or SVs), rather than using STIX index of 1KG / SGDP reads (see below for more details).

Does STIX improve the classification of pathogenic variants? The correlation with STIX and pathogenicity should be done statistically. For example, the authors could show that PCAWG / COSMIC variants that survive the STIX filter are enriched in Sanger cancer gene census genes. Similarly, for sick probands likely harboring a rare or de novo pathogenic variant vs healthy controls, one could show that STIX pass vs fail SVs are enriched among disease genes (eg via Clinvar, HGMD, genomeron) or in genes subject to purifying selection (eg LOFTEE, pLOF).

- The matched normal analysis results (Figure 2G) are encouraging but seem to be a missed opportunity to demonstrate the utility of the proposed filter. Perhaps the analyses is not clearly presented or it may be incorrectly formulated. It is also unclear why this analysis is limited to deletions, and not the full spectrum of cancer variants.

What is the gold standard call set in this analysis? I'm particularly confused how a FP / TP / FN rate is computable for the matched normal. The methods (STIX germline filtering evaluation) and figure caption were not informative here.

Stepping back, the premise of this analysis is to compare STIX/SGDP/1KG filtering with matched-normal filtering and standard "call set" filtering for 1KG and gnomAD. So the matched normal should be the gold standard, here. Granted a somatic SV caller (eg MANTA) may have its own false positives (which are the conclusions of the PCAWG analysis) but this small issue could be addressed in a number of ways, including querying a STIX index of the matched normal.

Encouragingly, it does appear that STIX removes many more variants than 1KG and gnomad, providing results that are in the ballpark of the matched-normal approach. But are these same SVs as the matched normal? It is not clear from this analysis.

The exclusive focus on DELs in this analysis is neither readily apparent nor justified in the text. Seems like it would be straightforward to extend this analysis to all SVs?

- Fig 2A-F are meant to convey the frequency and support (as a function of reads, samples) of false positives in COSMIC and PCAWG and their distribution in 1KG / SGDP. The (still overplotted scatter) plots in Fig 2A-F do not really bring this message home. For example, the origin presumably contains all the true positive variants, but its density is not apparent. It's also very small, and so it's very hard to visually assess what fraction of COSMIC / PCAWG variants have zero vs nonzero STIX support. It's also not visually apparent what (fraction of the) STIX hits are 1KG calls in these panels.

The authors may want to use density plots or add marginal densities to each axis. Bivariate plots may not be the ideal visual construct here. For example, Fig 2AB,DE it's not clear whether the relationship

between max read depth and num samples is important. If not, then univariate violin plots comparing various groups may be more evocative.

- Many (the majority in fact) of STIX hits are absent 1KG and SGDP callsets. Are most of these alignment artifacts or do these published analyses (drastically) under-call polymorphisms? My guess is the former i.e. STIX provides a powerful panel of normals for removing alignment artifacts in discordant pair analyses. This fact isn't emphasized much, so maybe I am misunderstanding? If not, then it should be more central to the message.

- The Tmprss2-ERG analysis provide a demo of the breakend region query but not clear how it contributes to the study practically or conceptually. The analysis is not mentioned anywhere in the text and neither is Supp Fig 1. The analysis is also set up in a fundamentally different way than the other analyses - namely it builds a STIX index of tumor reads rather than 1KG / SGDP and thus the goal is to confirm rather than refute a particular variant pattern. (Or perhaps it's just querying tumor SV breakend calls? If so, then it seems like something that could just be done via bedtools)?

In the rebuttal the authors seem to argue that STIX could be used to increase sensitivity (eg as a fast co-caller) rather than just the increasing the specificity of SV calling. Co-calling is generally not useful in cancer, unless analyzing clonally related samples from the same patient. Of the 130 cases where the fusion is found, how many were missed by other (single sample) callers? Perhaps a "comparative STIX analysis" of a low pass WGS cancer cohort vs SGDP/1KG could uncover SV drivers, but that seems out of the scope of this paper. The authors should better position this analysis with the main theme of the paper or exclude it.

Minor issues

- Figure captions should ideally just describe the graphic elements needed to visually understand the plot (eg each point represents, scatter plots of .. histograms of ..), but only currently only give high level interpretations of the plots. In many cases, it is not straightforward to connect the graphics to the captions, and vice versa. eg in 2A what is max per sample evidence depth, what is the red line in 2G. After some staring I was able to do some imputation, but it would help clarity dramatically if these were readily apparent.

- For groups of related analyses (eg Fig 2) it would be better if the panels have similar axes. For example 2G the different axes actually obscure the difference between the various filtering strategies.

- "When an inherited SV is missed in the normal tissue, it can be incorrectly classified as somatic" I think this statement misses a key utility of STIX, which is to uncover recurrent alignment artifacts. This is a key

benefit over using callsets which only contain population polymorphisms. This seems like a major point that is not sufficiently emphasized - unless I'm confused.

- "The SVs found by STIX are likely either germline or recurrent mutations and are unlikely to be driving tumor evolution." I don't understand this sentence. Again the STIX index should include alignment artifacts, unless I'm wrong? Also what are recurrent and non-germline mutations? STIX should help you distinguish de novo and somatic mutations from germline polymorphisms and alignment artifacts.

- Supplementary Figs 1 and 2 do not seem mentioned anywhere?

- Figure 2G "postivities"  positives

- Supplement Figure  Supplementary Figure (p15)

Reviewer #3:

Remarks to the Author:

1) Regarded read coverage, it looks like this was added:

"Another limitation is that STIX does not track per-sample normal coverage levels (due to high storage cost), which can mask the presence of deletions."

STIX is correctly positioned as a tool for searching large collections of genomes, and you are applying it broadly with SVs from curated callsets. I think your experiments are convincing, but I expect many will also apply STIX to their own callsets as a QC metric especially to support somatic or de novo variants in their own cohorts, which may be significantly smaller. How large must a cohort such that stochastic fluctuations do not significantly affect the results? Clearly, it depends on the experiment, coverage, and how rare variants might be. For good experimental design, it is really important to understand that 0 reads in matched somatic or parental samples could either mean no information or no SV support.

In my experience, bioinformaticians often miss this point leading to backtracking or incorrect results. I bring this point up again because based on what I have seen with STIX, I am certain it will be misapplied. To be clear, I don't think the method is flawed, but I do think you need do a little more to make sure it is properly applied.

Here is one suggestion:

"Since STIX does not track per-sample alignment depth, zero read support may result from no SV support or insufficient data at the locus. While STIX is designed for a large reference cohort where

alignment depth fluctuations in individual samples minimally impact the results, quantifying read depth or applying other QC metrics is advisable for smaller cohorts or particularly sensitive experiments involving rare variants."

Ways users can mitigate the limitation:

- a) Use a reference cohort large enough or further QC rare variants, especially important ones.
- b) Generate a read depth index and query it alongside STIX. I think 100-500 bp resolution would be fine for most cases.
- c) Proper QC, especially consider confidently callable regions using filters by GIAB or UCSC (e.g. tandem repeats and segmental duplications).

This is too much information for the manuscript itself, but please consider these ideas for your documentation.

Additional minor comments.

- 1) "Somatic false positives caused by false negatives in the control tissue are widespread". Certainly true, but do you have anything to back it up? A citation, an example?
- 2) Fig 1A, correct wording in "(A) A small of the alignments that tile the genomes are discordant"
- 3) There are many parenthetical statements inserted into sentences with parentheses or commas. It's purely style, but I think restructuring these sentences would improve clarity.

Author Rebuttal, first revision:

Reviewer #1:

Remarks to the Author:

I believe that the tool provided will be useful for variant filtering. The authors have addressed most of my comments from the previous round of review and adjusted their statements appropriately.

As STIX only uses insert size distribution and not the underlying sequence its ability to distinguish between similar SVs is more limited than many other short read SV callers that use the underlying sequence information. Using sequence information inside the STIX framework would be infeasible, but the statement that all other short read SV callers have this limitation is incorrect and needs to be adjusted.

STIX uses the insert size distribution to identify discordant read pairs and split reads, which depend on the underlying sequence information. Most short-read SV callers depend on these two signals, which is why we included that statement. The reviewer is correct that some SV callers go beyond discordant and split-reads, and we have changed that statement to be more specific.

I appreciate that these are not the first authors to do so, but in a scientific communication the abbreviation K should not be used to represent kilo.

The reviewer is absolutely correct. We chose to use "K" to be consistent with prior abbreviations.

Reviewer #2:

Remarks to the Author:

Pedersen et al have revised their work STIX, which introduces a novel web based index tool to enable fast structural variant filtering using publicly available databases, including 1KG and SGDP.

The key benefit of the STIX web service is to assess read-level evidence for candidate SVs in large databases. In general, the goal of that query is to identify (and remove) SV candidates that are common polymorphisms or alignment artifacts – and thus increase the specificity of SV calling. The main use cases are (1) clinical geneticists evaluating a (rare or de novo) constitutional variant (2) cancer genomicists wanting to improve (somatic) SV identification with or without a matched normal. The key premise of the work is that variants that survive the STIX filter are more likely to be pathogenic i.e. cause cancer or constitutional disease.

During revision, they have extended STIX to enable arbitrary breakend queries and also benchmarked their analysis for "tumor only" calling of structural variants. They have not supported the main claim of the paper related to pathogenicity. Their rebuttal also suggests many of the suggested revisions to be out of the scope of the current work.

Per the reviewer's suggestion, we added a new analysis that addresses the pathogenicity question using an experiment similar to what was suggested. Using VEP (McLaren et al., Genome Biology 2016) annotations, we showed that STIX filtering enriches the prostate cancer tumor call for SVs that are predicted to affect gene function (i.e. annotated as HIGH by VEP) versus SVs that are not predicted to affect a gene (annotated as MODIFIER by VEP) at about the same rate as matched-normal filtering (Figure below). We added this figure to the manuscript.

Figure. The density of VEP annotation types in tumor tissue calls, somatic calls that incorporated normal tissue, and tumor calls filtered using the 1KG STIX index. SVs that are predicted to affect gene function were annotated as HIGH, and those that don't were annotated as MODIFIER. The average per-sample number of SVs annotated as MODIFIER and HIGH in the tumor, tumor/normal, and STIX-filtered calls were 735.0 and 47.5, 28.6 and 22.8, and 10.0 and 10.5, respectively.

Nevertheless, just as not all somatic and rare SVs are pathogenic, we appreciate the reviewer's point that not all SVs predicted to affect gene function are pathogenic. The advantage that STIX provides is to reduce the number of the SVs that would be considered for disease or cancer progression. We have made these points in the manuscript.

Despite the new functionality and analytic updates, the manuscript still does not convey the technical advance and utility, in my view. Some of this may be an issue with data presentation. I do think that will stand in the way of wide adoption.

We appreciate the reviewer's point, but respectfully disagree for two primary reasons. First, the novel technical advance and utility of STIX is in its ability to reduce the thousands of SVs identified by an SV caller (e.g., MANTA) to, on average, fewer than 50 variants that are both somatic and functionally relevant (Figure 2G). Just as gnomAD drastically improved our ability to interpret SNVs by allowing us to filter variants based on population frequency estimates, STIX can do the same for SVs.

Secondly, we emphasize the importance of the fact that our results show that unlike SNV lists (e.g., gnomAD), SV lists from 1KG and gnomAD SV underperform in filtering while STIX-based filtering rivals matched normal filters, which are the current gold-standard. By making STIX available as a web service, any SV disease project can quickly and easily reduce their analysis burden and improve their results. By making the underlying software open source, institutions can create their own local STIX indexes, which would be particularly interesting to clinical applications and would also address privacy concerns. The latest version of the manuscript highlights these points.

Specific concerns

- The authors have not addressed the correlation of STIX with pathogenicity, which is central to the paper (part of the title) but barely examined.

As discussed above, we performed an analysis on the PCAWG tumor genomes that showed that STIX-filtered SV call sets are enriched for SVs that are predicted to affect gene function. We agree with the reviewer's point that "pathogenic" is not the appropriate label to use. However, while these SVs are not necessarily pathogenic, they are putative consequential. We thank the reviewer for raising this point; the latest version of the manuscript discusses this distinction.

Their rebuttal to this critique refers to an analysis (see below) that addresses the ability of STIX to detect somatic variants. Most somatic variants however are not pathogenic ie are not cancer drivers. Similarly, de novo variants may also not be pathogenic.

The reviewer is correct. While population frequency is the most powerful metric available to the genomics community for pathogenic variant identification, not all rare variants (somatic or de novo) are pathogenic and classifying a variant as pathogenic requires additional analysis. Furthermore, interpreting the effect of an SV on gene function or regulation is still an open question. Claiming that STIX itself identifies pathogenic SVs is not as precise as claiming that STIX enriches call sets for SVs that are predicted to affect gene function. We have added this as a limitation of the approach in the newest version of the manuscript.

The TMPRSS2-ERG analysis is included a supplementary figure but not referred to in the text (neither is the figure). It is also confusing - set up in a fundamentally different than the main application of STIX the paper. Namely, it builds a STIX index of cancer reads (or SVs), rather than using STIX index of 1KG / SGDP reads (see below for more details).

There was a paragraph that described the analysis and referenced the figures on page 4. We agree that this analysis is orthogonal to the germline population frequency analysis. However, we also believe that STIX can play a critical role in understanding disease genomic SV frequency, just as it can play a critical role in germline SV frequency. We have improved the logical flow of the manuscript by moving this discussion to the end, after the germline filtering analysis.

Does STIX improve the classification of pathogenic variants? The correlation with STIX and pathogenicity should be done statistically. For example, the authors could show that PCAWG / COSMIC variants that survive the STIX filter are enriched in Sanger cancer gene census genes. Similarly, for sick probands likely harboring a rare or de novo pathogenic variant vs healthy controls, one could show that STIX pass vs fail SVs are enriched among disease genes (eg via Clinvar, HGMD, genomeron) or in genes subject to purifying selection (eg LOFTEE, pLOF).

This is a very good suggestion and we have added an analysis similar to the one suggested above. Using VEP (McLaren et al., Genome Biology 2016) , we annotate all of the SVs found by MANTA in the PCAWG prostate tumor samples, then tracked the rate at which STIX removed SV that were predicted to affect gene function (annotated as HIGH) and those that were not (annotated as MODIFIER). STIX-filtered call sets were enriched with gene altering SVs at the same rate as call sets filtered using calls from matched normal samples (Figure above). We added this figure to the manuscript.

- The matched normal analysis results (Figure 2G) are encouraging but seem to be a missed opportunity to demonstrate the utility of the proposed filter. Perhaps the analyses is not clearly presented or it may be incorrectly formulated. It is also unclear why this analysis is limited to deletions, and not the full spectrum of cancer variants.

The utility we intend to provide with STIX is to allow users to refine their SV callsets down from thousands of variants to a manageable set that can be reasonably curated by hand. This is the same utility that calls from matched normal tissue provide in cancer studies and calls from parents in Mendelian disease studies. The difference being that with STIX the filtering does not require these additional samples, which are not always available. We have added text addressing this utility to the manuscript.

We have extended our analysis to include duplications and inversions.

What is the gold standard call set in this analysis? I'm particularly confused how a FP / TP / FN rate is computable for the matched normal. The methods (STIX germline filtering evaluation) and figure caption were not informative here.

The published ICGC/PCAWG class was the truth set. False positives (FP) were the SVs that passed the filters but were not in the PCAWG calls. True positives (TP) were the SVs that passed the filters and were in the PCAWG calls. False negatives (FN) were SVs that did not pass the filters and were in the PCAWG calls. We have modified the main text and methods to make this clear.

Stepping back, the premise of this analysis is to compare STIX/SGDP/1KG filtering with matched-normal filtering and standard "call set" filtering for 1KG and gnomAD. So the matched normal should be the gold standard, here. Granted a somatic SV caller (eg MANTA) may have its own false positives (which are the conclusions of the PCAWG analysis) but this small issue could be addressed in a number of ways, including querying a STIX index of the matched normal.

Picking a gold standard in this context is difficult. The objective of this analysis was to measure how well STIX would perform in a tumor analysis. That is, how close are STIX-filtered tumor-only calls to a final call set. In this context we believe that the final call set for PCAWG, which included extra filters to remove false positives (as the reviewer pointed out), is a better comparison point than the MANTA matched/normal calls. The MANTA results for the matched/normal calls are included in the results from which the reader can still compare it to STIX.

Encouragingly, it does appear that STIX removes many more variants than 1KG and gnomad, providing results that are in the ballpark of the matched-normal approach. But are these same SVs as the matched normal? It is not clear from this analysis.

Yes. 99.9% of the STIX true positives are also MANTA tumor/normal true positives. As reflected in Figure 2G, the Manta tumor/normal finds, on average, 6.7 more true positives per sample.

The exclusive focus on DELs in this analysis is neither readily apparent nor justified in the text. Seems like it would be straightforward to extend this analysis to all SVs?

We have extended our analysis to include duplications and inversions.

- Fig 2A-F are meant to convey the frequency and support (as a function of reads, samples) of false positives in COSMIC and PCAWG and their distribution in 1KG / SGDP. The (still overplotted scatter) plots in Fig 2A-F do not really bring this message home. For example, the origin presumably contains all the true positive variants, but its density is not apparent. It's also very small, and so it's very hard to visually assess what fraction of COSMIC / PCAWG variants have zero vs nonzero STIX support. It's also not visually apparent what (fraction of the) STIX hits are 1KG calls in these panels.

Figures 2A-F are meant to convey that the SVs in these somatic SV databases appear across the spectrum of both per-sample depths and allele frequency, and that SV lists miss SVs that also span the allele frequency spectrum. We agree that the figures are small, which was done to meet the format requirements. Increasing the size of the figure would require removing some, and in our opinion the latter option is worse. As suggested by the reviewer, we added what fraction of the hits are recovered in the legend. This fraction is also useful in understanding what fraction of COSMIC / PCAWG variants have zero vs non-zero STIX support.

The authors may want to use density plots or add marginal densities to each axis. Bivariate plots may not be the ideal visual construct here. For example, Fig 2AB,DE it's not clear whether the relationship between max read depth and num samples is important. If not, then univariate violin plots comparing various groups may be more evocative.

In our opinion the relationship between the evidence depth and number of samples is important in understanding the likelihood of false positives. For example, if the SVs that STIX identified as being at high frequency also had exclusively low sample depth it may be interpreted that STIX only recovered the low-level random noise and not signal for a polymorphic SV.

- Many (the majority in fact) of STIX hits are absent 1KG and SGDP callsets. Are most of these alignment artifacts or do these published analyses (drastically) under-call polymorphisms? My guess is the former i.e. STIX provides a powerful panel of normals for removing alignment artifacts in discordant pair analyses. This fact isn't emphasized much, so maybe I am misunderstanding? If not, then it should be more central to the message.

The current version does state that SVs found by STIX are either "common germline variants or the product of systematic noise." We have added alignment artifacts as an example of systematic noise.

- The TMPRSS2-ERG analysis provide a demo of the breakend region query but not clear how it contributes to the study practically or conceptually. The analysis is not mentioned anywhere in the text and neither is Supp Fig 1. The analysis is also set up in a fundamentally different way than the other analyses - namely it builds a STIX index of tumor reads rather than 1KG / SGDP and thus the goal is to confirm rather than refute a particular variant pattern. (Or perhaps it's just querying tumor SV breakend calls? If so, then it seems like something that could just be done via bedtools)?

There is a paragraph that describes this analysis with references to the figures on page 4. The object is to demonstrate the flexibility of STIX in both indexing a different population (in this case prostate cancers) and submitting a different type of query for a different type of analysis (in this case a query for a "functionally equivalent" SV). To summarize the analysis, we show that a "standard" STIX query that recovered evidence for a particular SV (in this case a particular TMPRSS2-ERG fusion) in only a small number of other prostate cancer patients, which is surprising given how common these fusions are in prostate cancer.

In the rebuttal the authors seem to argue that STIX could be used to increase sensitivity (eg as a fast co-caller) rather than just the increasing the specificity of SV calling. Co-calling is generally not useful in cancer, unless analyzing clonally related samples from the same patient. Of the 130 cases where the fusion is found, how many were missed by other (single sample) callers? Perhaps a "comparative STIX analysis" of a low pass WGS cancer cohort vs SGDP/1KG could uncover SV drivers, but that seems out of the scope of this paper. The authors should better position this analysis with the main theme of the paper or exclude it.

The idea behind co-calling was born out of a collaboration with oncologists who are performing deep targeted sequencing and had an interest in differentiating between true low-frequency events and noise. While we agree that rediscovering a specific SV is much more likely in related samples, a large reference panel could be useful in recovering systematic noise. This is clearly speculation and would require, as the reviewer suggests, experiments.

There was an issue in our last result. The number of samples that STIX identified as having a fusion event was 129 (not 130). This error is now fixed in the latest version. PCAWG identified fusions in 105 samples. All of these 105 samples were also found by STIX.

We agree that the analysis is out of place. In the latest version of the paper we moved it after all of the experiments that used the SGDP/1KG indexes.

Minor issues

- Figure captions should ideally just describe the graphic elements needed to visually understand the plot (eg each point represents, scatter plots of .. histograms of ..), but only currently only give high level interpretations of the plots. In many cases, it is not straightforward to connect the graphics to the captions, and vice versa. eg in 2A what is max per sample evidence depth, what is the red line in 2G. After some staring I was able to do some imputation, but it would help clarity dramatically if these were readily apparent.

We have improved the captions per these suggestions.

- For groups of related analyses (eg Fig 2) it would be better if the panels have similar axes. For example 2G the different axes actually obscure the difference between the various filtering strategies.

We agree and we spent many hours trying to get the axes on the same scale. Unfortunately the differences between ranges condense the STIX and tumor/normal bars such that they are very difficult to interpret.

- "When an inherited SV is missed in the normal tissue, it can be incorrectly classified as somatic" I think this statement misses a key utility of STIX, which is to uncover recurrent alignment artifacts. This is a key benefit over using callsets which only contain population polymorphisms. This seems like a major point that is not sufficiently emphasized - unless I'm confused.

It could be the case that STIX removes mostly recurrent alignment artifacts. It could also be the case that STIX is recovering real SVs that are missed or filtered by SV callers. As we get more long-read data we may be able to determine which of these is more common. Either way, the key utility of STIX is to help identify novel SVs, which we think we have made more clear in our title and latest revision.

- "The SVs found by STIX are likely either germline or recurrent mutations and are unlikely to be driving tumor evolution." I don't understand this sentence. Again the STIX index should include alignment artifacts, unless I'm wrong? Also what are recurrent and non-germline mutations? STIX should help you distinguish de novo and somatic mutations from germline polymorphisms and alignment artifacts.

We wanted to account for the possibility that the SVs that STIX finds occurred independently in different samples. We added text about alignment artifacts.

- Supplementary Figs 1 and 2 do not seem mentioned anywhere?

Supplementary Figure 1 is referenced on page 4, but we accidentally remove the Supplementary Figure 2 reference. Based on order of appearance, the order of these two figures is now switched in the newest version.

- Figure 2G "postivities"  positives

Fixed. Thank you.

- Supplement Figure  Supplementary Figure (p15)

Fixed. Thank you.

Reviewer #3:

Remarks to the Author:

1) Regarded read coverage, it looks like this was added:

"Another limitation is that STIX does not track per-sample normal coverage levels (due to high storage cost), which can mask the presence of deletions."

STIX is correctly positioned as a tool for searching large collections of genomes, and you are applying it broadly with SVs from curated callsets. I think your experiments are convincing, but I expect many will also apply STIX to their own callsets as a QC metric especially to support somatic or de novo variants in their own cohorts, which may be significantly smaller. How large must a cohort such that stochastic fluctuations do not significantly affect the results? Clearly, it depends on the experiment, coverage, and how rare variants might be. For good experimental design, it is really important to understand that 0 reads in matched somatic or parental samples could either mean no information or no SV support.

In my experience, bioinformaticians often miss this point leading to backtracking or incorrect results. I bring this point up again because based on what I have seen with STIX, I am certain it will be misapplied. To be clear, I don't think the method is flawed, but I do think you need do a little more to make sure it is properly applied.

Here is one suggestion:

"Since STIX does not track per-sample alignment depth, zero read support may result from no SV support or insufficient data at the locus. While STIX is designed for a large reference cohort where alignment depth fluctuations in individual samples minimally impact the results, quantifying read depth or applying other QC metrics is advisable for smaller cohorts or particularly sensitive experiments involving rare variants."

We completely agree with this suggestion and have added a slightly edited version of this paragraph to our manuscript.

Ways users can mitigate the limitation:

- a) Use a reference cohort large enough or further QC rare variants, especially important ones.
- b) Generate a read depth index and query it alongside STIX. I think 100-500 bp resolution would be fine for most cases.
- c) Proper QC, especially consider confidently callable regions using filters by GIAB or UCSC (e.g. tandem repeats and segmental duplications).

This is too much information for the manuscript itself, but please consider these ideas for your documentation.

We will add these suggestions and the above warning to our GitHub documentation. Thank you.

Additional minor comments.

1) "Somatic false positives caused by false negatives in the control tissue are widespread". Certainly true, but do you have anything to back it up? A citation, an example?

Good point. Our results show that false positives are wide-spread, but we are unable to definitely tie them to false negatives in the normal. We removed that sentence in the current version.

2) Fig 1A, correct wording in "(A) A small of the alignments that tile the genomes are discordant"

The wording is now fixed. Thank you.

3) There are many parenthetical statements inserted into sentences with parentheses or commas. It's purely style, but I think restructuring these sentences would improve clarity.

We have gone through the manuscript and cleaned many of these issues.

Decision Letter, second revision:

Subject: AIP Decision on Manuscript NMETH-BC45904B

Message:

Our ref: NMETH-BC45904B

14th Dec 2021

Dear Dr. Layer,

Thank you for submitting your revised manuscript "Searching Thousands of Genomes to Classify Somatic and Novel Structural Variants" (NMETH-BC45904B). It has now been seen by Reviewer 2 and their comments are below. After evaluating the review report and discussion within the editorial team, we'll be happy in principle to publish it in Nature Methods, pending revisions to satisfy the referees' final requests and to comply with our editorial and formatting guidelines.

In light of Reviewer 2's review report, we suggest you either exclude the Tmprss2-ERG analysis, or move it to Supplementary Information and tone down (eg, by acknowledging weakness of the analysis

and result interpretation as raised by Reviewer 2). Meanwhile, please further improve figure captions (eg by defining axes in figures).

TRANSPARENT PEER REVIEW

Nature Methods offers a transparent peer review option for new original research manuscripts submitted from 17th February 2021. We encourage increased transparency in peer review by publishing the reviewer comments, author rebuttal letters and editorial decision letters if the authors agree. Such peer review material is made available as a supplementary peer review file. Please state in the cover letter 'I wish to participate in transparent peer review' if you want to opt in, or 'I do not wish to participate in transparent peer review' if you don't. Failure to state your preference will result in delays in accepting your manuscript for publication.

Thank you again for your interest in Nature Methods Please do not hesitate to contact me if you have any questions.

Sincerely,

Lin Tang, PhD
Senior Editor
Nature Methods

ORCID

Reviewer #2 (Remarks to the Author):

Chowdhury et al present a revised manuscript describing STIX. While the edits have improved the manuscript somewhat, it is disappointing to see continuing lack of clarity in key analyses after so many revisions, particularly around points that were already addressed in previous reviews.

- The TMPRSS2-ERG fusion analysis still does not make sense. It also seems to be built on some basic misconceptions about somatic structural variation. The use case is also not well described and seems a bit contrived.

Even highly recurrent SVs (like TMPRSS2-ERG, BCR-ABL) are associated with a broad range of genomic DNA breaks since the breaks occur intronically. So it is completely expected that any given breakpoint combination would be rarely seen, even at for TMPRSS2-ERG.

The rebuttal refers to oncology - can the authors spell out the scenario here? i.e. you have a patient sample with some candidate rearrangement. Now you have a STIX index eg of PCAWG or of all the samples you have sequenced in the clinical lab.

If you find many unrelated patient samples with the identical genomic breakpoint, that is most likely a germline or artifact.

The range query also has unclear utility. So now you expand your query around the candidate SV e.g. to the gene level, and find that there are many hits in the STIX tumor DB. Is that significant? Perhaps if your population DB had few hits in the same query, then you would may think that this an SV that is under selection and perhaps is more likely to be important for cancer. However that comparison is not described.

But if this is the case, then why wouldn't a standard query of a bedpe of called tumor SVs find the same hits and uncover the same pattern? Perhaps these SVs are subclonal in most of your reference tumors and never made it to the final callset in each reference tumor sample. Or maybe they are artifacts, for example related to batch (e.g. FFPE).

Fundamentally a STIX tumor DB query is not telling you about the analytic validity of a call in a brand new sample, but may tell you about evolutionary selection. However for that sort of analysis it is unclear how STIX provides benefits over querying the somatic calls themselves.

- "which is far less than is expected" - this is a bit of a straw man. Again, tumor SVs, even those that cause recurrent fusion drivers, have unique genomic breakpoint locations even when they result in the same fusion transcript. There is no selection within the intron and even mutational biases do not create

hotspots at the breakpoint level. So again, it is completely expected that any given genomic breakpoint combination would be rarely seen, even for TMPRSS2-ERG.

- How do the 129 samples compare to the PCAWG gold standard calls? Currently the section makes it seem like the query is uncovering previously undetected variants. My guess is that these are concordant - should be mentioned.

- "we used STIX to investigate .. a standard STIX query" From the rebuttal, the index appears to be built from tumor reads, which is different than the previous examples. Up until this point in the manuscript, STIX has been used to query reference population dbs of normal human blood. It should be made explicit here what is the query and what is the db, as mentioned in the last round.

- "reveal .. mutational mechanisms driving a tumor's rearranged genomes". Unclear what is being revealed about mechanisms. The location shown here seems to be driven by fusion exon structure which is actually under selection. If the distribution validates observations in the cited study, the comparison should be made more explicit.

- Captions have not improved. As noted before, the captions provide interpretation but do not describe what is being shown. The captions should describe the data. This greatly limits the clarity of the manuscript, particularly to a new reader.

Minor comments:

- typo: "This query also recapitulated a previously observed location bias ERG"

Author Rebuttal, second revision:

Response to Reviewer 2

Reviewer #2:

Remarks to the Author:

Chowdhury et al present a revised manuscript describing STIX. While the edits have improved the manuscript somewhat, it is disappointing to see continuing lack of clarity in key analyses after so many revisions, particularly around points that were already addressed in previous reviews.

1. The TMPRSS2-ERG fusion analysis still does not make sense. It also seems to be built on some basic misconceptions about somatic structural variation. The use case is also not well described and seems a bit contrived.

Even highly recurrent SVs (like TMPRSS2-ERG, BCR-ABL) are associated with a broad range of genomic DNA breaks since the breaks occur intronically. So it is completely expected that any given breakpoint combination would be rarely seen, even at for TMPRSS2-ERG.

The rebuttal refers to oncology - can the authors spell out the scenario here? i.e. you have a patient sample with some candidate rearrangement. Now you have a STIX index eg of PCAWG or of all the samples you have sequenced in the clinical lab.

If you find many unrelated patient samples with the identical genomic breakpoint, that is most likely a germline or artifact.

The range query also has unclear utility. So now you expand your query around the candidate SV e.g. to the gene level, and find that there are many hits in the STIX tumor DB. Is that significant? Perhaps if your population DB had few hits in the same query, then you would may think that this an SV that is under selection and perhaps is more likely to be important for cancer. However that comparison is not described.

But if this is the case, then why wouldn't a standard query of a bedpe of called tumor SVs find the same hits and uncover the same pattern? Perhaps these SVs are subclonal in most of your reference tumors and never made it to the final callset in each reference tumor sample. Or maybe they are artifacts, for example related to batch (e.g. FFPE).

Fundamentally a STIX tumor DB query is not telling you about the analytic validity of a call in a brand new sample, but may tell you about evolutionary selection. However for that sort of analysis it is unclear how STIX provides benefits over querying the somatic calls themselves.

2. "which is far less than is expected" - this is a bit of a straw man. Again, tumor SVs, even those that cause recurrent fusion drivers, have unique genomic breakpoint locations even when they result in the same fusion transcript. There is no selection within the intron and even mutational biases do not create hotspots at the breakpoint level. So again, it is completely expected that any given genomic breakpoint combination would be rarely seen, even for TMPRSS2-ERG.

3. How do the 129 samples compare to the PCAWG gold standard calls? Currently the section makes it seem like the query is uncovering previously undetected variants. My guess is that these are concordant - should be mentioned.

4. "we used STIX to investigate .. a standard STIX query" From the rebuttal, the index appears to be built from tumor reads, which is different than the previous examples. Up until this point in the manuscript, STIX has been used to query reference population dbs of normal human blood. It should be made explicit here what is the query and what is the db, as mentioned in the last round.

4. "reveal .. mutational mechanisms driving a tumor's rearranged genomes". Unclear what is being revealed about mechanisms. The location shown here seems to be driven by fusion exon structure which is actually under selection. If the distribution validates observations in the cited study, the comparison should be made more explicit.

Response to points 1-4:

In light of the reviewers remaining issues with the TMPRSS2-ERG fusion analysis, we have chosen to remove it from the manuscript. Further work will need to be done in order to strengthen this analysis for a potential standalone submission.

5. Captions have not improved. As noted before, the captions provide interpretation but do not describe what is being shown. The captions should describe the data. This greatly limits the clarity of the manuscript, particularly to a new reader.

We apologize for the continued confusion with the figure captions and have rewritten the caption for figure 2 to both include a high level interpretation as well as more granular description of figure axes/data points to make it easier for a new reader to interpret.

Minor comments:

- typo: "This query also recapitulated a previously observed location bias ERG"
This section has now been removed.

Final Decision Letter:

Subject: Decision on Nature Methods submission NMETH-BC45904C
Message:

13th Feb 2022

Dear Dr Layer,

I am pleased to inform you that your Brief Communication, "Searching Thousands of Genomes to Classify Somatic and Novel Structural Variants using STIX", has now been accepted for publication in Nature Methods. Your paper is tentatively scheduled for publication in our April print issue, and will be published online prior to that. The received and accepted dates will be 23rd Apr 2021 and 13th Feb 2022. This note is intended to let you know what to expect from us over the next month or so, and to let you know where to address any further questions.

Your paper will now be copyedited to ensure that it conforms to Nature Methods style. Once proofs are generated, they will be sent to you electronically and you will be asked to send a corrected version within 24 hours. It is extremely important that you let us know now whether you will be difficult to contact over the next month. If this is the case, we ask that you send us the contact information (email, phone and fax) of someone who will be able to check the proofs and deal with any last-minute problems.

If, when you receive your proof, you cannot meet the deadline, please inform us at rjsproduction@springernature.com immediately.

Once your manuscript is typeset and you have completed the appropriate grant of rights, you will receive a link to your electronic proof via email with a request to make any corrections within 48 hours. If, when you receive your proof, you cannot meet this deadline, please inform us at rjsproduction@springernature.com immediately.

Once your paper has been scheduled for online publication, the Nature press office will be in touch to confirm the details.

Content is published online weekly on Mondays and Thursdays, and the embargo is set at 16:00 London time (GMT)/11:00 am US Eastern time (EST) on the day of publication. If you need to know the exact publication date or when the news embargo will be lifted, please contact our press office after you have submitted your proof corrections. Now is the time to inform your Public Relations or Press Office about your paper, as they might be interested in promoting its publication. This will allow them time to

prepare an accurate and satisfactory press release. Include your manuscript tracking number NMETH-BC45904C and the name of the journal, which they will need when they contact our office.

About one week before your paper is published online, we shall be distributing a press release to news organizations worldwide, which may include details of your work. We are happy for your institution or funding agency to prepare its own press release, but it must mention the embargo date and Nature Methods. Our Press Office will contact you closer to the time of publication, but if you or your Press Office have any inquiries in the meantime, please contact press@nature.com.

Please note that *Nature Methods* is a Transformative Journal (TJ). Authors may publish their research with us through the traditional subscription access route or make their paper immediately open access through payment of an article-processing charge (APC). Authors will not be required to make a final decision about access to their article until it has been accepted. [Find out more about Transformative Journals](https://www.springernature.com/gp/open-research/transformative-journals)

Authors may need to take specific actions to achieve [compliance](https://www.springernature.com/gp/open-research/funding/policy-compliance-faqs) with funder and institutional open access mandates. For submissions from January 2021, if your research is supported by a funder that requires immediate open access (e.g. according to [Plan S principles](https://www.springernature.com/gp/open-research/plan-s-compliance)) then you should select the gold OA route, and we will direct you to the compliant route where possible. For authors selecting the subscription publication route our standard licensing terms will need to be accepted, including our [self-archiving policies](https://www.springernature.com/gp/open-research/policies/journal-policies). Those standard licensing terms will supersede any other terms that the author or any third party may assert apply to any version of the manuscript.

To assist our authors in disseminating their research to the broader community, our SharedIt initiative provides you with a unique shareable link that will allow anyone (with or without a subscription) to read the published article. Recipients of the link with a subscription will also be able to download and print the PDF. As soon as your article is published, you will receive an automated email with your shareable link.

Please note that you and your coauthors may order reprints and single copies of the issue containing your article through Springer Nature Limited's reprint website, which is located at

<http://www.nature.com/reprints/author-reprints.html>. If there are any questions about reprints please send an email to author-reprints@nature.com and someone will assist you.

Best regards,

Lin Tang, PhD
Senior Editor
Nature Methods